# Structural biases in disordered proteins are prevalent in the cell

**David Moses** [1,2,10], **Karina Guadalupe** [1,2,10], **Feng Yu**[2,3], **Eduardo Flores**[1,2], **Anthony R. Perez**[1,2], **Ralph McAnelly**[1], **Nora M. Shamoon**[2,4], **Gagandeep Kaur**[1], **Estefania Cuevas-Zepeda** [1], **Andrea D. Merg**[1,2], **Erik W. Martin** [5,9], **Alex S. Holehouse** [6,7] **& Shahar Sukenik** [1,2,3,8] ✉

Intrinsically disordered proteins and protein regions (IDPs) are prevalent in all proteomes and are essential to cellular function. Unlike folded proteins, IDPs exist in an ensemble of dissimilar conformations. Despite this structural plasticity, intramolecular interactions create sequence-specific structural biases that determine an IDP ensemble's three-dimensional shape. Such structural biases can be key to IDP function and are often measured in vitro, but whether those biases are preserved inside the cell is unclear. Here we show that structural biases in IDP ensembles found in vitro are recapitulated inside human-derived cells. We further reveal that structural biases can change in a sequence-dependent manner due to changes in the intracellular milieu, subcellular localization, and intramolecular interactions with tethered well-folded domains. We propose that the structural sensitivity of IDP ensembles can be leveraged for biological function, can be the underlying cause of IDP-driven pathology or can be used to design disorder-based biosensors and actuators.

Intrinsically disordered proteins and protein regions (IDPs) play key roles in many cellular pathways and are vital to cellular function in all kingdoms of life[1]. Compared to folded proteins, IDPs lack a stable tertiary structure, have fewer intramolecular interactions, and expose a greater area of their sequence to the surrounding solution[2]. As a result, an IDP exists in an ensemble of highly dissimilar conformations that can change rapidly in response to the physical–chemical characteristics of its surroundings[3].

Despite being highly dynamic, IDP ensembles often contain structural biases, or preferences for certain subsets of conformations within the ensemble[4]. Such structural biases may arise from short- or long-range interactions within the protein sequence (Fig. 1a)[5]. An extensive body of work has established the importance of IDP ensemble structure to their function. For example, local biases that form transient α-helical segments modulate binding affinity in PUMA[6] and p53 (ref. 7) and the liquid–liquid phase separation properties of TDP-43 (ref. 8). Changes to long-range structural biases were found to influence IDP function in p53 (ref. 9), BMAL1 (ref. 10) and Myc[11]. However, with few exceptions[12–14], studies linking IDP ensemble structure to function are performed in vitro. The differences between an aqueous buffer and the cellular environment are dramatic[15], casting doubt as to whether or not structural biases linked to function in vitro persist in the cell.

The structural malleability of IDP ensembles, coupled with the dynamic nature of the cellular environment, prompts two major unanswered questions: (1) To what degree are IDP structural biases observed in vitro preserved inside the cell? (2) How do IDP structural

[1]Department of Chemistry and Biochemistry, University of California, Merced, Merced, CA, USA. [2]Center for Cellular and Biomolecular Machines, University of California, Merced, Merced, CA, USA. [3]Quantitative and Systems Biology Program, University of California, Merced, CA, USA. [4]California State University, Stanislaus, Turlock, CA, USA. [5]Department of Structural Biology, St Jude Children's Research Hospital, Memphis, TN, USA. [6]Department of Biochemistry and Molecular Biophysics, Washington University School of Medicine, St. Louis, MO, USA. [7]Center for Science and Engineering of Living Systems, Washington University in St. Louis, St. Louis, MO, USA. [8]Health Sciences Research Institute, University of California, Merced, Merced, CA, USA. [9]Present address: Dewpoint Therapeutics, Boston, MA, USA. [10]These authors contributed equally: David Moses, Karina Guadalupe. ✉e-mail: ssukenik@ucmerced.edu

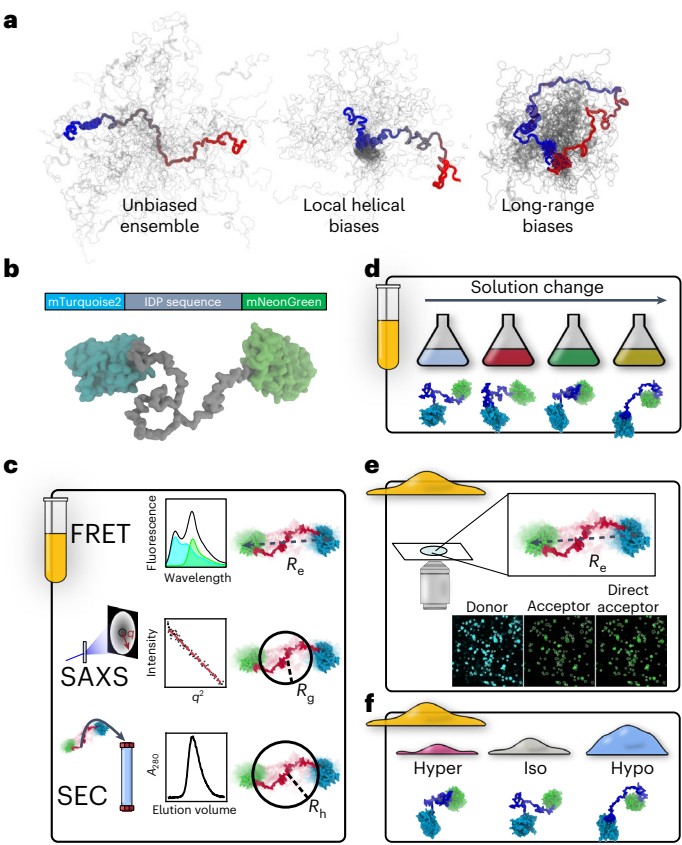

**Fig. 1 | Methods to compare in vitro and in-cell IDP ensembles. a**, IDP ensembles with and without structural biases. In all schemes, a single conformation is shown in color and other conformations are shown in gray. Structural biases increase the density in specific regions of the ensemble and alter its average dimensions. **b**, FRET construct consisting of an IDP between two FPs that serve as a FRET donor and a FRET acceptor. **c**, In vitro experiments. Top: FRET. Middle: SAXS. Bottom: analytical SEC. **d**, In vitro solution space scanning measures the sensitivity of ensemble structure to changes in solution conditions. **e**, Live cell FRET microscopy is performed on cells expressing the same constructs used in vitro. **f**, Changes in ensemble dimensions are measured using live cell FRET following rapid hyperosmotic and hypoosmotic challenges. Hyper, hyperosmotic; Iso, isoosmotic; Hypo, hypoosmotic.

biases respond to physical–chemical changes in the dynamic intracellular environment?

To answer these questions requires a correlative approach that combines both in vitro and live cell studies. We have established a characterization pipeline that combines ensemble fluorescence resonance energy transfer (FRET) (Fig. 1b,c), analytical size-exclusion chromatography (SEC) (Fig. 1c), small angle X-ray scattering (SAXS) (Fig. 1c), changes in solution composition (Fig. 1d), and molecular simulations to identify structural biases of IDPs in vitro. We then examine the same constructs inside live cells using FRET microscopy (Fig. 1e). Finally, we perturb the cellular ensembles by subjecting cells to osmotic challenges that rapidly change cell volume and measure the response of IDP ensembles through changes in FRET signal (Fig. 1f).

In this Article, using this approach, we find that the structural biases that define IDP ensembles in vitro also exist inside the cell. Furthermore, we highlight cases where IDPs respond in a sequence-dependent manner to osmotic challenges, changes in subcellular localization or interaction with a folded domain. Our results demonstrate that IDP structural biases can be tuned by changes to protein sequence or to the cellular environment.

## Glycine–serine repeats are an unbiased, model-free standard

The structure of a folded protein is commonly described in terms of its 'native' conformation discerned through X-ray crystallography. For an IDP, no single structure can be obtained. Instead, IDP structure is often described with reference to well-established homopolymer models[16,17]. However, no models exist for our dumbbell-shaped FRET construct (Fig. 1b), especially not models that are relevant in the cellular environment. We therefore wanted to create an empirical standard against which we could compare IDPs of arbitrary lengths in different environments.

As a benchmark against which to compare properties of naturally occurring heteropolymeric IDPs, we inserted homopolymeric dipeptide repeats into our FRET construct (Supplementary Data 1). Specifically, we chose glycine–serine (GS) repeats for benchmarking, because (1) they lack hydrophobicity, charge and aromaticity, which makes them easy to express and highly soluble[4], (2) they have been shown to lack local and long-range structural biases, instead behaving as expected for a random coil across the range of lengths studied in our work[18], and (3) they have been shown to behave as real-chain mimics of ideal Gaussian chains in aqueous solutions[18,19].

Ensemble FRET experiments provide an apparent FRET efficiency ($E_f^{app}$), which is inversely proportional to the distance between the two fluorescent proteins (FPs) in our FRET construct. When the FPs are close together, $E_f^{app}$ is high, and when they are far apart, $E_f^{app}$ is low, indicating compaction or expansion of an ensemble. As previously reported, $E_f^{app}$ decreased linearly with the number of GS repeats in a dilute buffer solution[3] (Fig. 2a,b and Supplementary Fig. 1).

To obtain additional information about the three-dimensional structure of the ensemble, we performed SEC coupled with SAXS (SEC–SAXS) on the constructs we had measured using FRET. The chromatograms obtained from SEC showed a consistent, linear, size-dependent increase in elution volume (Fig. 2c,d and Supplementary Fig. 2), indicating that the proteins increase in dimension with GS repeat length. Analysis of SAXS intensity curves showed a similar linear dependence on GS length (Fig. 2e,f and Supplementary Figs. 3 and 4), displaying linearly increasing radii of gyration ($R_g$; Fig. 2f) in agreement with our other results.

Finally, we conducted all-atom simulations of all GS repeat sequences to enable a molecular benchmark between SAXS and FRET results. Our simulations assumed that the FPs are non-interacting and that GS repeats behave like ideal homopolymers. Ensembles we selected from these simulations to quantitatively match our SAXS scattering data (Supplementary Fig. 5) reproduced our GS length-dependent $E_f^{app}$ values as well, showing consistency between our orthogonal FRET and SAXS results (Fig. 2b,f).

Taken together, our methods consistently show the same length-dependent trend for the GS repeats, and that the length of the sequence is the dominant factor affecting these dimensions. The excellent quantitative agreement with our simulations further confirms that GS repeats behave like ideal homopolymers, which lack structural biases.

To further verify that GS repeats do not contain structural biases, we conducted FRET-based solution space scanning of GS repeat constructs[3,20]. Solution space scanning measures the change in FRET efficiency ($\Delta E_f^{app} = E_{f,solute}^{app} - E_{f,buffer}^{app}$). $\Delta E_f^{app}$ probes structural biases in the ensemble by modulating interactions between the sequence and the solution. We reason that if structural biases exist, different GS repeat lengths will show a different structural response to the same solution. As expected, GS repeats of all lengths responded identically to each of the solution conditions we created (Supplementary Fig. 6). Overall, the internal consistency of the results from our orthogonal characterization methods establishes GS repeats as a model-free homopolymer standard, which lacks structural biases.

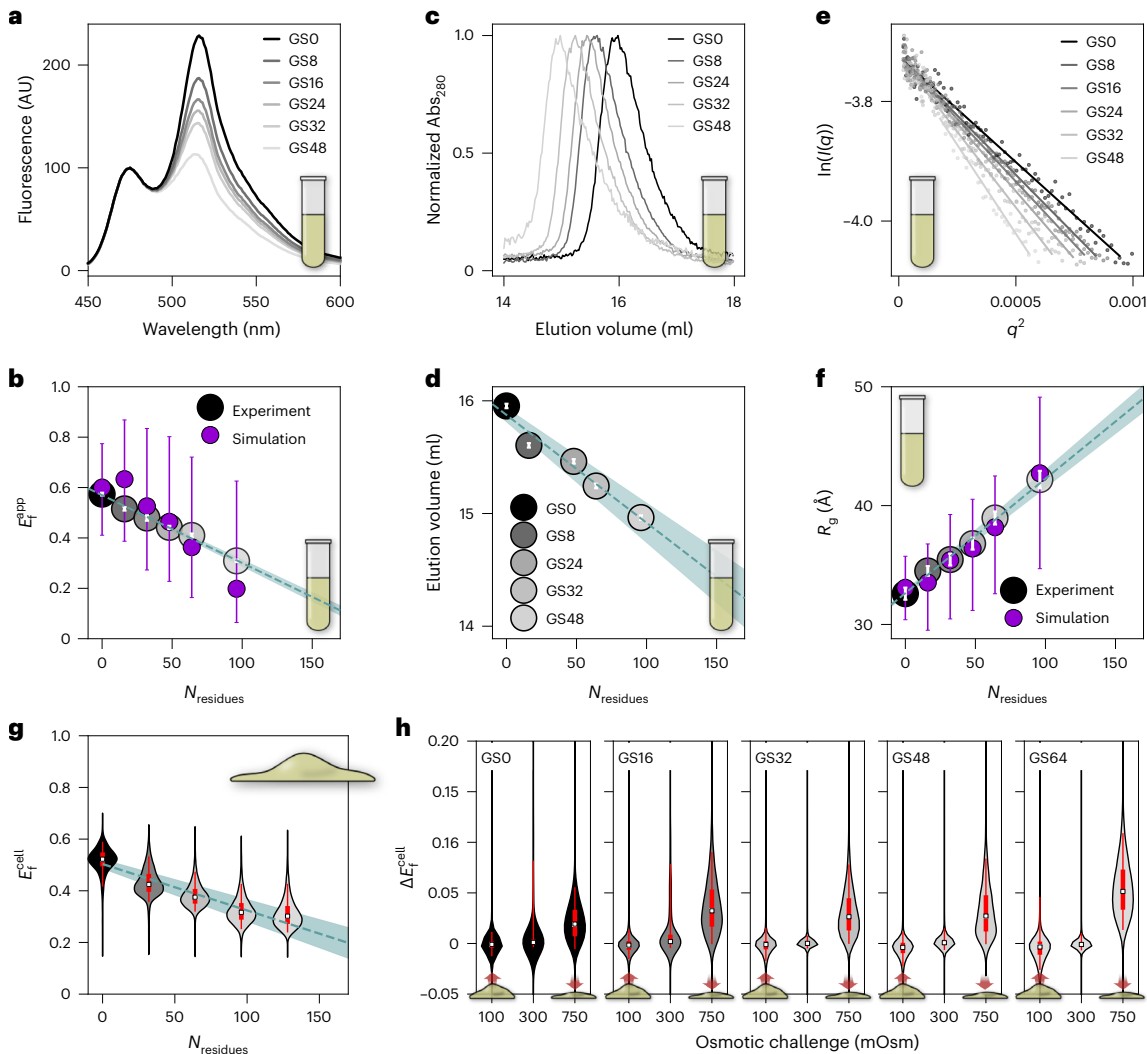

**Fig. 2 | Characterization of GS repeat standards. a**, Fluorescence spectra from in vitro measurements of FRET GS*X* constructs, where *X* indicates the number of GS repeats. **b**, Average apparent in vitro FRET efficiencies ($E_f^{app}$) of GS repeats. Error bars represent the standard deviation ($N = 12$). Purple circles indicate all-atom simulation calculations of $E_f^{app}$ (Supplementary Fig. 5) and error bars represent the median 50% of the simulated ensemble. Dashed teal line shows a linear fit of the averages with errors shown as the shaded teal region. **c**, SEC chromatograms for GS repeats. **d**, SEC elution volumes, expressed as the position of the peak in ml versus number of residues in the GS repeat sequence. Errors are obtained from determination of peak position in Supplementary Fig. 2. Dashed teal lines as in **b**. **e**, Guinier regions and linear fits from SAXS experiments for GS repeats. **f**, Radii of gyration ($R_g$) derived from Guinier analysis of SAXS data for GS repeats. White error bars represent errors from linear fitting of Guinier plots. Dashed teal lines and purple markers are the same as in **b**. **g**, FRET efficiencies of GS repeats measured in live cells ($E_f^{cell}$). In all live cell results, violin plots span the entire dataset and their thickness represents $E_f^{cell}$ probability. For each violin, the median is shown as a white square; thick and thin red lines are the median 50% and 95% of the data, respectively. Dashed teal line shows a linear fit of the medians, and fit error is shown by the shaded region. **h**, $\Delta E_f^{cell}$ for GS repeats. Features are as in **g**. The dataset used to generate all of the live cell figures is in Supplementary Data 3. *N* for each violin plot is in Supplementary Data 4.

## Live cell measurements recapitulate in vitro GS repeat results

We next sought to establish GS repeats as a bias-free standard in live cells. To facilitate direct and straightforward comparison with our in vitro experiments, we used the same genetically encoded FRET constructs as we had used in vitro. GS repeat FRET constructs were expressed in HEK293T cells, which all showed similar morphology and expression levels regardless of the construct being expressed (Supplementary Fig. 7).

Our live cell measurements of GS repeats showed trends in FRET efficiency calculated from live cell imaging ($E_f^{cell}$) that are in quantitative agreement with in vitro measurements (Fig. 2b,g). Notably, in live cells our FRET constructs showed a much broader distribution of $E_f^{cell}$ compared with the distribution of $E_f^{app}$ shown in vitro. This variability may

be caused by a range of factors, including cell-to-cell differences in composition, cell state and construct expression levels. Despite this, the remarkable agreement with in vitro data indicates that the lack of structural biases for GS repeats detected in vitro persists inside live cells.

To test whether GS ensemble dimensions are sensitive to the cellular environment, we subjected cells to osmotic challenge. To resolve their immediate effects on a protein, these perturbations are performed rapidly and measured as quickly as possible to prevent any kind of transcriptional response[21,22]. We use rapid osmotic challenges induced by the addition of NaCl (hyperosmotic, to a final 750 mOsm) or water (hypoosmotic, to a final 100 mOsm) to Dulbecco's modified Eagle medium (DMEM) (which is isosmotic at 300 mOsm). Osmotic challenges were previously shown to produce robust and reproducible

changes in cellular volume through the efflux or influx of water[21–23]. We report on the difference in FRET signal of each cell following this perturbation, $\Delta E_f^{cell} = E_{f,after}^{cell} - E_{f,before}^{cell}$ (Fig. 2h). The measurements before and after the challenge are collected within a span of 45 s or less.

Hyperosmotic perturbations resulting in cell shrinkage caused a positive $\Delta E_f^{cell}$ that scaled with the length of the construct (Fig. 2h and Supplementary Fig. 8). This is in line with previous studies of IDPs in crowded conditions and in the cell[17] and can be explained by the increased ability of longer sequences to compact. Hypoosmotic perturbations, on the other hand, produced no substantial change in $E_f^{cell}$ (Supplementary Fig. 8). This lack of response was surprising, especially since GS polymers are capable of expansion in vitro (Supplementary Fig. 6). Regardless, our osmotic challenge experiments define a standard for the response of bias-free IDP ensembles to osmotically induced changes in cellular volume.

## Amino acid sequence determines IDP structural biases

Having established a reliable homopolymer standard in vitro and in live cells, we set out to investigate how a naturally occurring IDP compares with GS repeats. We chose the sequence of the PUMA BH3 domain (wild-type (WT) PUMA) (Fig. 3a,b and Supplementary Data 1) because its residual helicity is a well-studied example of functionally linked structural biases in IDPs[6,24]. We first established the previously reported short-range helical structural biases of the unlabeled sequence[25] as indicated by the characteristic double minima in the circular dichroism (CD) spectrum (Fig. 3b,c and Supplementary Fig. 9). Next, we measured the $E_f^{app}$, $R_g$ and SEC elution volume of WT PUMA using our in vitro pipeline (WT in Fig. 3d–f). Although in SEC WT PUMA eluted near the same volume as would be expected of GS repeats of the same length (Fig. 3e), SAXS and FRET showed WT PUMA to be substantially more compact than corresponding GS repeats (Fig. 3d,f), confirming that we are able to detect local structural biases present in WT PUMA but absent in GS repeats.

Is residual helicity similar to that observed in WT PUMA a prerequisite for detectable structural biases? To answer this question, we generated sequence scrambles of WT PUMA (Fig. 3a and Supplementary Data 1) and measured their ensembles in vitro. Sequence scrambles retain the amino acid composition but change their order, disrupting structural biases present in the WT. The three scrambles of WT PUMA were designed to have varying degrees of charge clustering in the sequence (sequences S1–3; Fig. 3a,b). To test for the existence of helical structural biases in the scrambled sequences, we measured the secondary structure of the label-free IDPs using CD. As expected, the CD spectra of the scrambles showed no double minima (Fig. 3c and Supplementary Fig. 9), indicating that the helical structural biases of WT PUMA were no longer present.

We next characterized ensemble dimensions of the scrambles using FRET (Fig. 3d), SEC (Fig. 3e), SAXS (Fig. 3f) and all-atom Monte Carlo simulations (Supplementary Fig. 10). FRET and SAXS show that

not only are the scrambles more compact than GS repeats of the same length, but they also all differ from each other despite having similar CD spectra and identical amino acid composition (Fig. 3a–c). The overall agreement between trends from FRET and SAXS measurements shows that the WT PUMA ensemble is the most compact, followed by S2, S3 and finally S1. This trend is recapitulated in label-free all-atom simulations, indicating that tethering to the two FP labels does not change the trends in ensemble dimensions for this measurement (Supplementary Fig. 10). SEC data show a different trend, with all sequences appearing more expanded than a GS linker and S3 showing an almost equal compaction to WT (Fig. 3e). This may be due to chemical interactions between the constructs and the SEC column matrix[26]. However, since all four sequences contain the same amino acid composition, even these different interactions indicate sequence-dependent structuring within the ensemble.

The differences shown by all methods between WT PUMA and the three scrambles demonstrate not only that the WT PUMA ensemble is uniquely more compact than the scrambles, but also that structural biases exist even in the absence of the helical structural biases in the WT sequence. These results also show that, in this case, charge patterning alone does not dictate ensemble dimensions, since S3 has similar patterning to WT but is substantially more expanded according to FRET and SAXS results.

We hypothesized that different structural biases in WT PUMA and the scrambles would also manifest in their response to different solutions. To test this, we performed solution space scans for all four PUMA variants (Fig. 3h and Supplementary Fig. 11). We compare $\Delta E_f^{app}$ of each sequence against the interpolated $\Delta E_f^{app}$ of GS repeats of the same length in the same solution condition (Fig. 3h and Supplementary Fig. 12). Deviations from $\Delta E_f^{app}$ of length-equivalent GS repeats indicate higher/lower sensitivities of the sequences (indicated by red/blue backgrounds, respectively) (Fig. 3h). We were surprised to find that, despite having the most compact ensemble, WT PUMA showed the highest sensitivity of all scrambles. Specifically, the WT sequence displayed stronger compaction in response to polymeric crowders (specifically PEG2000) and stronger expansion in response to denaturants (urea and GuHCl) than both the corresponding GS repeat sequence and the three sequence scrambles. The three scrambles showed milder responses, with S2 especially insensitive to all solutes. These differences indicate that IDPs possess sequence-encoded sensitivity to the chemical composition of their environment. Furthermore, the presence of structural biases does not preclude ensemble sensitivity to the surrounding solution, and may even amplify it.

## Sequence-dependent structural biases persist in live cells

We next wanted to see if the structural biases measured in vitro for WT PUMA and its scrambles were retained inside the cell. We expected helical structural biases to persist in the cell due to the intrinsic stability of this secondary structure[27], but reasoned that biases within the

**Fig. 3 | Sequence-dependent structural biases of PUMA BH3 domain. a**, Sequence of WT PUMA BH3 domain and three sequences (S1, S2 and S3) derived by shuffling WT PUMA's sequence. $\kappa$ measures clustering of charged residues in the sequence, with a value closer to 1 for sequences where like charges are highly clustered. **b**, Molecular features of PUMA sequences. FCR, fraction of charged residues; NCPR, net charge per residue; Hydro: Kyte–Doolittle hydrophobicity. Values are the average of the five nearest residues. **c**, CD spectroscopy of PUMA variants without flanking FPs. See also Supplementary Fig. 9. **d**, Average $E_f^{app}$ of PUMA constructs. Error bars represent the standard deviation ($N = 12$), the teal dashed line is the interpolated value for a 34-residue GS repeat, and the shaded teal area is the error from the interpolation. Blue/red shading indicates expansion/compaction compared to GS repeat. **e**, SEC elution volume for PUMA constructs. Errors are obtained from determination of peak position in Supplementary Fig. 2. Teal dashed line and blue/red shading as in **d**.

**f**, SAXS-derived $R_g$ of PUMA constructs. Error bars are calculated from linear fits to Guinier plots (Supplementary Fig. 3). Teal dashed line and blue/red shading as in **d**. **g**, $E_f^{cell}$ of PUMA constructs. Features are as in Fig. 2g. Teal dashed line and blue/red shading as in **d**. **h**, $\Delta E_f^{app}$ of PUMA constructs in response to solution changes. Black dashed lines are interpolated $\Delta E_f^{app}$ of a GS repeat sequence of the same length as the IDP (Supplementary Fig. 12). Green shaded regions are differences between $\Delta E_f^{app}$ of IDP and GS repeats. Gray error bars indicate the spread of the data over two repeats. The background color for each plot indicates the sensitivity of the IDP to that solute, with red/blue being more/less sensitive (compared to the GS repeat). **i**, $\Delta E_f^{cell}$ of PUMA constructs (violins) and GS repeat equivalents (squares). Features are as in Fig. 2g. The dataset used to generate all live cell figures is in Supplementary Data 3. $N$ for each violin plot is in Supplementary Data 4.

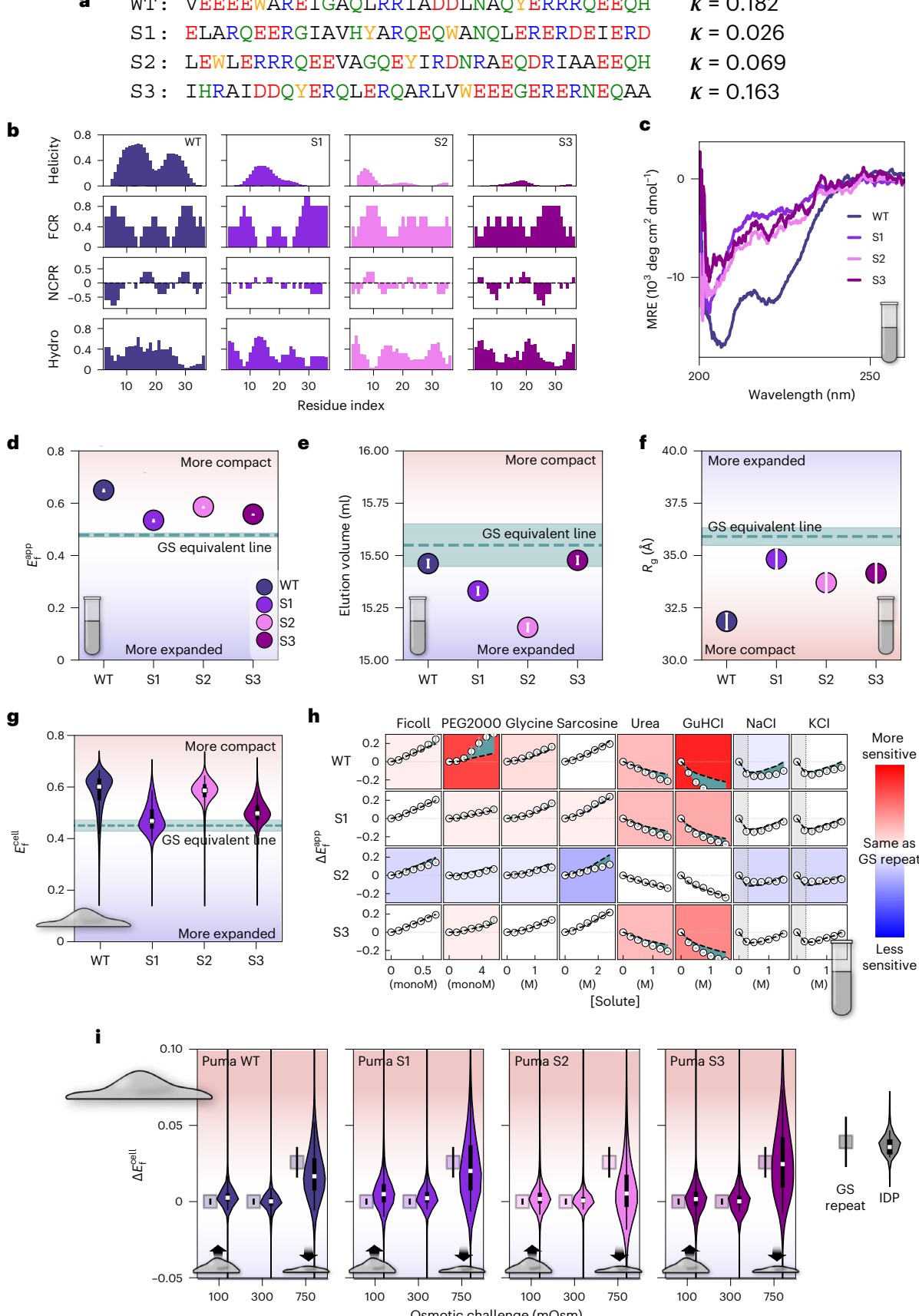

scrambled sequences were weaker and therefore might not be retained. To test this, we performed our live cell FRET imaging experiments on WT PUMA and the three scrambles (Fig. 3g). Our live cell FRET experiments showed striking agreement with the FRET measurements done in dilute aqueous buffers (Fig. 3d). Specifically, both the relative magnitude and the trend in $E_f^{app}$ measured in vitro was replicated in live cells, with WT > S2 > S3 > S1. Overall, $E_f^{cell}$ reveals that the structural biases found in these sequences in vitro persist inside the cell, even in the absence of short-range helical structural biases (which occur only in WT; Fig. 3c).

Our next goal was to measure whether these ensembles differ in their response to changes in the cellular environment. We again used osmotically triggered cell volume perturbations as a means to reproducibly change the concentration of all cellular solutes. $\Delta E_f^{cell}$ is reported and compared to the interpolated $\Delta E_f^{cell}$ for GS repeats of the same length (Fig. 3i). We were surprised to find that the WT sequence, which displayed more sensitivity than corresponding GS repeats to certain solutes in vitro, showed a response similar to that of GS repeats under both cell volume increase and decrease. Remarkably, this similarity to GS repeat sensitivity in live cells was seen in all sequences except S2, which displayed a markedly lower tendency to compact under hyperosmotic conditions (as indicated by the lack of overlap between the median 50% of the data and the GS repeat equivalent). The lower sensitivity of S2 was also observed in vitro (Fig. 3h). This result indicates that IDP ensemble sensitivity to changes in the cellular environment is encoded in sequence, but is difficult to predict since it may or may not correlate with the sensitivity measured in dilute buffers.

## Biases in naturally occurring IDPs persist inside the cell

Having seen that structural biases in vitro persist inside the cell for PUMA and its scrambles, we wanted to see whether this is a general property of other IDP sequences. We inserted a range of well-studied naturally occurring IDPs of different lengths into our construct and characterized them in vitro and in live cells. We tested the N-terminal disordered region of p53 (residues 1–61, p53)[7], which contains the N-terminal activation domain[7], the low-complexity domain of FUS (residues 1–163, FUS)[28], the N-terminal region of the adenovirus hub protein E1A (residues 1–40, E1A)[29], and the C-terminal region of the yeast transcription factor Ash1 (residues 418–500, Ash1)[30] (Supplementary Fig. 13 and Supplementary Data 1). Importantly, the ensemble structure of each of these IDPs has previously been characterized in vitro and has been shown or proposed to determine IDP function (Discussion).

Using our in vitro characterization pipeline, we found clear divergence in nearly all constructs from GS repeats. Our FRET experiments show that three sequences (PUMA, E1A and FUS) are more compact than a GS repeat sequence of the same dimensions (Fig. 4a). The two that fell close to the GS line, p53 and Ash1, have been reported to be relatively expanded in other studies[7,30]. A similar trend was observed for SAXS-derived $R_g$ values (Fig. 4c). SEC data (Fig. 4b) show mostly similar trends, although PUMA, E1A and p53 appear to be more expanded than GS repeats. As before, the deviations from the GS-equivalent line, together with the changes in trends between characterization methods, highlight the differences in structural biases between different IDP sequences.

Our next goal was to determine the extent to which the structural biases observed in vitro for these constructs persist in the cell. Using live cell imaging to quantify $E_f^{cell}$, we observe good agreement between $E_f^{app}$ measured in vitro and the $E_f^{cell}$ values (Fig. 4a,d). As before, this agreement indicates that structural biases that determine IDP ensemble shape in vitro largely exist inside the cell.

We next wanted to see how the localization of IDPs in the cell might affect their ensembles. We reasoned that different organelles have different physical–chemical compositions, and this may affect the ensemble preferences encoded in IDP sequences[31]. To test this idea, we measured $E_f^{cell}$ in the cytoplasm and nucleus of U-2 OS cells for all our sequences. GS repeats showed the same $E_f^{cell}$ in both cytoplasm and nucleus within error, indicating their ensemble is unaffected by changes in localization (Supplementary Fig. 14). All $E_f^{cell}$ measurements were normalized to a GS repeat of the same length (Fig. 4e). Most sequences showed no substantial difference between the cytoplasm and the nucleus. An exception was observed for the FUS low-complexity domain, which was more expanded in the nucleus (Fig. 4e). This might be due to its ability to interact with nuclear-abundant RNA[32].

## Naturally occurring IDPs differ in solution sensitivity

Next, we performed solution space scanning on PUMA, FUS, p53, Ash1 and E1A (Fig. 4f and Supplementary Fig. 15). As expected, different sequences showed markedly different sensitivities to the solutes used. PUMA and Ash1 showed an outlying degree of sensitivity, with larger changes compared to GS repeats of the same length in both compacting and expanding solutes, while E1A appeared to be less sensitive to the same solutes (Fig. 4f). The response to salts also showed deviations, with less response to high salt concentrations for E1A. Interestingly, p53, whose dimensions were closest to those of its GS equivalent in dilute buffer (Fig. 4a), also displayed sensitivity most similar to its GS equivalent (Fig. 4f). In line with our previous results[3], we found that PEG2000 produces greater increases in $E_f^{app}$ than the smaller PEG400 at equal monomer–molar concentrations, and that the monomer units of the crowders (sucrose and ethylene glycol) produce relatively small changes in the dimensions of the IDPs. This wide range of responses to changes in solution conditions further supports the existence of sequence-dependent structural biases found in our FRET, SAXS and SEC results. Moreover, the different IDP ensembles show differing and specific sensitivities to changes in their chemical environment.

Finally, we wanted to measure the response of these IDPs to changes in intracellular composition. We subjected cells to hypoosmotic or hyperosmotic challenges and followed the changes in average FRET signal for each cell, $\Delta E_f^{cell}$ (Fig. 4g). We compare these to the changes expected for GS repeats of the same length, shown as the squares adjacent to each violin plot. We found that PUMA, Ash1, FUS and p53 all fell within the range expected of their GS repeat equivalents. FUS displayed a similar behavior to GS repeats upon hyperosmotic challenge, but showed an outlying ability compared to the other naturally occurring IDPs to expand in hypoosmotic conditions. However, most striking was E1A's response to cellular perturbations. Expansion of IDPs under increased crowding has been previously reported in vitro[33] and may be caused inside the cell by protein–protein interactions such as chaperone binding[34] or post-translational modifications[35].

Taken together, these results show not only that structural biases in IDP ensembles exist both in vitro and inside the cell, but also that IDP ensembles are able to sense and respond to changes in the composition of their environment. This ability is encoded in sequence and occurs both in the test tube and in the cell. However, despite the agreement between IDP structural biases in a dilute solution in vitro and in isosmotic conditions in the cell, comparing in vitro and in-cell solution sensitivity is not straightforward.

## Interactions between IDPs and their tethered folded domains

One alternative possibility that could explain the aberrant behavior of E1A is that the IDP interacts intramolecularly with one or both of the FPs in our FRET construct, and that cellular perturbations disrupt this interaction. To test whether IDP ensemble structural biases are influenced by interactions with the tethered FPs, we repeated our FRET experiments using constructs with the locations of the FPs flipped from their original locations (Fig. 5a). We reasoned that since the surface of each FP (Supplementary Fig. 16a), their termini (Supplementary

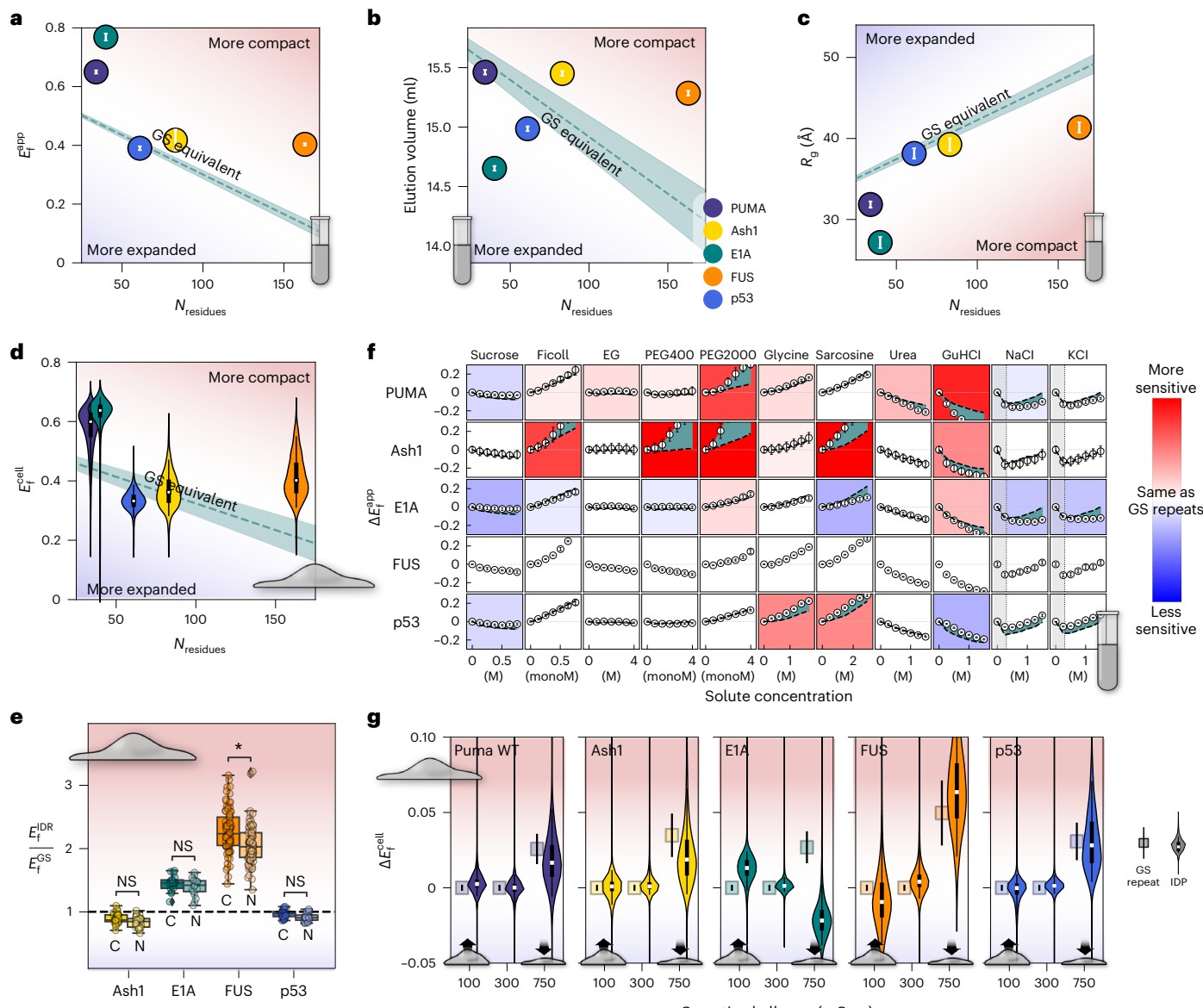

**Fig. 4 | Comparison of global dimensions and solution sensitivity of GS repeats and naturally occurring IDPs. a**, Average $E_f^{app}$ of IDP constructs. Error bars represent standard deviation from $N = 12$ repeats. Teal dashed line represents the same as in Fig. 2b. Blue/red shading indicates expansion/compaction compared to GS repeat. **b**, SEC elution volume of IDP constructs. Errors are obtained from determination of peak position in Supplementary Fig. 2. Teal dashed line represents the same as in Fig. 2d. Blue/red shading as in **a**. **c**, $R_g$ of IDP constructs. Errors are from fitting lines to Guinier plots. Teal dashed line represents the same as in Fig. 2f. Blue/red shading as in **a**. **d**, $E_f^{cell}$ of IDP constructs. Violin features are as in Fig. 2g. Blue/red shading as in **a**. **e**, $E_f^{cell}$ of four IDP

constructs measured in the cytoplasm (C) and nucleus (N) of U-2 OS cells and normalized to the $E_f^{cell}$ of an equivalent GS linker. The black horizontal line is the median, each box is the median 50% and the whiskers show the minimum and maximum of the data for each construct. Each point corresponds to a single cell. $P$ values were obtained from a two-sided Mann–Whitney Wilcoxon test (NS, no significance; *$P = 0.00006$). **f**, Solution space scans showing IDP constructs. Features are as in Fig. 3h. **g**, $\Delta E_f^{cell}$ of IDP constructs (violins) and the GS repeat equivalent (squares) are shown. Features are as in Fig. 2h. The dataset used to generate all of the live cell figures is in Supplementary Data 3. $N$ for each violin plot is in Supplementary Data 4.

Fig. 16b) and the termini of the IDP differ, changes in FRET signal in the flipped versus the original construct would indicate the involvement of interactions between the IDP and the FPs in determining $E_f^{app}$.

As with previous experiments, we started with a GS repeat sequence. In this case, the IDP termini are identical, and any difference would be a result of changes in the FPs themselves rather than a difference in IDP:FP interactions. Our in vitro measurements showed a higher $E_f^{app}$ for the flipped GS16 construct, indicating a more compact conformation (Fig. 5b). Further NaCl titration experiments and analysis of raw fluorescence spectra showed that (1) electrostatic interactions do

not account for the difference in $E_f^{app}$ (Supplementary Fig. 17a–c), and that (2) the difference in $E_f^{app}$ between the original and flipped construct is probably a result of changes in the structure of the mNeonGreen tail tethered to the IDP (Supplementary Fig. 17d). As described above, our analysis indicates that $E_f^{app}$ of GS repeat homopolymers is not driven by IDP:FP interactions. When measured in live cells, flipped GS16 again displayed similar results to those seen in vitro, with a higher $E_f^{cell}$ for the flipped GS16 construct (Fig. 5c).

We next compared the basal in vitro $E_f^{app}$ and live cell $E_f^{cell}$ distributions of the original and flipped versions of three previously measured

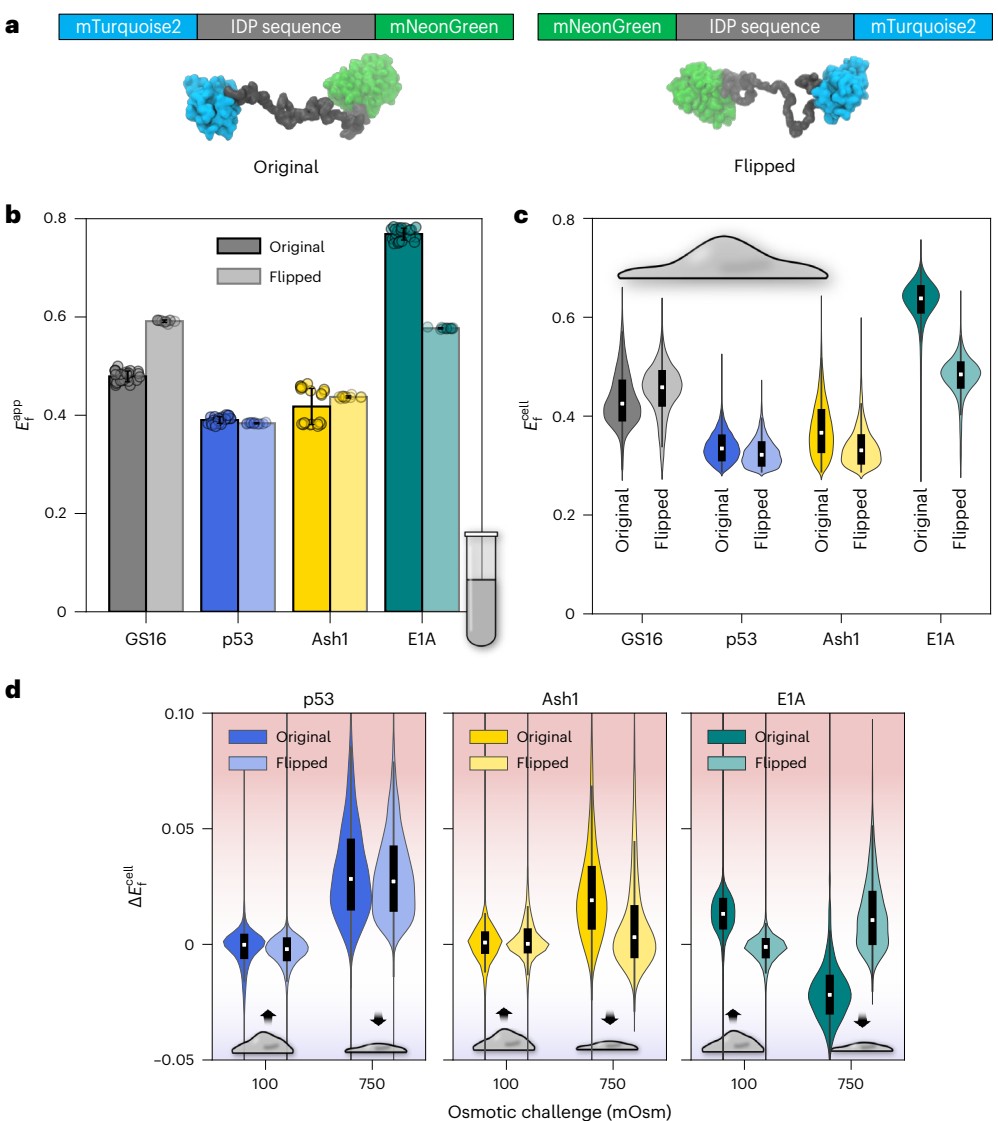

**Fig. 5 | Determination of FRET pair influence on IDP ensemble dimensions.**
**a**, Original FRET construct (left) consisting of an IDP between two FPs that serve as a FRET donor and a FRET acceptor and the flipped construct (right) with the FRET pairs on the opposite ends. **b**, $E_f^{app}$ of selected constructs measured in vitro for the original (darker color) and flipped (lighter color) constructs. $N = 24$ for original and $N = 12$ for flipped constructs. **c**, $E_f^{cell}$ of selected constructs measured in HEK293T cells for original (darker color) and flipped (lighter color) constructs. Features are as in Fig. 2g. **d**, $\Delta E_f^{cell}$ of original (darker color) and flipped (lighter color) constructs. Features are as in Fig. 2g,h. The dataset used to generate the live cell figures is in Supplementary Data 3. $N$ for each violin plot is in Supplementary Data 4.

constructs (Fig. 4g): E1A (whose original version had shown a different response than GS repeats to osmotic challenge), Ash1 (whose original version had only shown a difference in hyperosmotic conditions compared to GS repeats), and p53 (whose original version had shown a similar response to GS repeats to osmotic challenge). Unlike GS16, these naturally occurring IDPs contain different sequences at their N and C termini, as well as charged residues that could contribute to electrostatic interactions between the FPs and IDP. Both in vitro and in cells, E1A displayed a dramatic reduction in FRET efficiency, while flipped Ash1 and p53 showed little change compared to the original constructs (Fig. 5d). This points to interactions between one or both of the FPs and the sequence of E1A. As with GS16, further tests of emission peak wavelengths implicated mNeonGreen as the FP with substantial changes to its spectrum upon tethering. It also showed a different trend in peak wavelength shift for E1A compared to Ash1 and p53 (Supplementary Fig. 17d).

We hypothesized that if there were changes to the ensemble in the flipped construct, it would also alter the response to changes in cell volume. Testing this, we indeed found that p53, but not Ash1, displayed similar responses to changes in cell volume (Fig. 5e). This is despite p53 and Ash1 having similar dimensions between the original and flipped constructs. E1A, on the other hand, showed a completely opposite response between the flipped and original constructs (Fig. 5e). These results indicate that IDP:folded domain interactions can alter the ensemble's response to changes in the cellular environment. But regardless of these differences between the constructs, the ensemble dimensions as measured by FRET efficiency remain similar in vitro and in the cell.

## Limitations and drawbacks

One drawback of this work is the use of FPs in our constructs. There are many advantages to genetically encoded FRET constructs. They can be produced easily in *Escherichia coli* with no need for further labeling. They can also be transiently or stably expressed in any genetically tractable cell line and measured directly. Additionally, the FPs flanking the

sequence increase solubility and signal from scattering methods and hinder aggregation and phase separation.

However, as indicated for E1A, the presence of bulky folded domains tethered to the IDP of interest may affect our results through intramolecular interactions of the FPs with each other or with the IDP sequence. We acknowledge that interactions between the studied IDPs and the FPs that make up our FRET construct exist and probably affect the dimensions of our measured ensembles.

Nonetheless, concerns regarding artifacts from our use of FPs are mitigated by (1) the use of the same FPs for all constructs and the comparison against GS repeat constructs, which facilitate meaningful comparison between all sequences, and (2) the agreement between our experiments and all-atom simulations of the GS repeats (Fig. 1b,f and Supplementary Fig. 10). Also, our results show that even where FP:IDP interactions are seen to exist, the structural biases shaping disordered protein ensembles in vitro are recapitulated in the cell.

Finally, we note that nearly all studied IDPs (including those in this work) are excised from full-length proteins, in which they would be tethered to folded domains. The importance of IDP:folded domain interactions has already been pointed out in several recent studies[36,37]. Our results point to the importance of the intramolecular context of an IDP. Specifically, we show that interactions with a tethered folded domain can alter IDP ensembles, as well as their response to changes in the cell.

## Discussion

The study of disordered proteins requires shifting from the classical sequence–structure–function paradigm to one where the structural biases of the ensemble beget function[4]. While an extensive body of work has established the existence of structural biases in IDP ensembles in vitro, few studies have attempted to do so in the cell across many constructs in a self-consistent manner. Our results systematically show that structural biases are prevalent in IDP sequences, are encoded in amino acid sequence rather than composition, and exist even in the absence of local secondary structural biases (for example, local helical preference; Fig. 1a).

The cell is often treated as a chemically monolithic environment, yet spatial and temporal regulation of volume, water content, pH, ions and metabolites accompany key processes and pathology in cell biology[38,39]. Our in-cell study establishes that IDP structural biases observed in vitro also occur in live cells for almost all cases reported here. Furthermore, both in cells and in vitro, IDP structural biases can reshape in response to changes in the surrounding environment. This provides a mechanistic explanation for numerous cases where IDPs sense and actuate a response to such changes[40–42], since a change in structural bias in response to physical–chemical changes can alter IDP function. Importantly, sensing and actuating through this mechanism occurs at the speed of protein conformational changes (milliseconds or less[25]) and requires no additional energy (for example, ATP).

The importance of IDP ensembles for molecular function has been shown or proposed for all of the naturally occurring IDPs characterized in this study. The structural preferences of the PUMA BH3 ensemble have been shown to affect its binding kinetics to MCL1—a key event in the function of PUMA as a modulator of p53, and it has further been shown that this structural change can be induced by changing the composition of the solution[6]. Changing the structural preferences of the p53 N-terminal ensemble affects its binding affinity to MDM2, a potent inhibitor of p53's protective function, altering downstream p53 function[7]. FUS low-complexity region can undergo phase separation in vitro and in vivo[32]. The Ash1 ensemble has been shown to remain largely unperturbed by phosphorylation, indicating the need for robust activity of this yeast transcription factor[30]. Finally, a region proximal to the E1A sequence used here has been shown to be highly conserved in terms of the average end-to-end distance of its ensemble, and this length critical to its function, implicating strong selection for its ensemble dimensions[43].

Given that IDP ensemble sensitivity can be encoded by amino acid sequence, we suggest that this sensitivity could also be subject to evolutionary selection. We propose that certain sequences have evolved to act as sensors and actuators of changes in the cellular environment. This sensing capability of IDPs has been demonstrated not only for changes in solution conditions and osmotic pressure as studied here, but also for changes in other conditions such as membrane curvature[40], water availability[42] and temperature[44]. As our understanding of IDP sensing expands, we expect to uncover novel functions for this important class of proteins. In addition, learning to predict and control this sensitivity will allow for the design of IDP-based sensors targeting specific physicochemical intracellular conditions, as has already been demonstrated for the case of osmotic pressure sensing[42].

An additional implication of the evolved ability to sense and respond to changes in the environment is that a misregulated intracellular environment may disparately affect IDP function. Metabolic rewiring, a hallmark of cancer, viral infection and other pathologies, can dramatically alter the physicochemical composition of the cell[45]. Even if this change would alter the activity of only a small subset of IDPs, their role as central signaling hubs could cause widespread cellular malfunction. In this way, IDP sequences can be drivers of pathology in a deleterious cellular environment, even in the absence of mutations. We propose that this phenomenon is a previously overlooked cause of IDP-driven proteopathies.

## Online content

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

## Methods

### FRET construct design and cloning

The FRET backbone for bacterial expression (fIDP_pET-28a(+)-TEV) or for mammalian expression (fIDP_pCDNA3.1(+)) was prepared by ligating mTurquoise2 and mNeonGreen into pET28a-TEV or plasmid cloning DNA backbone using 5′ NdeI and 3′ XhoI restriction sites. Genes encoding for IDP regions were obtained from GenScript and ligated between the two FPs using 5′ SacI and 3′ HindIII restriction sites. Cloned plasmids were amplified in XL1 Blue cell lines (Thermo Fisher Scientific) using the manufacturer-supplied protocol. Sequences of all IDP inserts are available in Supplementary Data 1.

### FRET construct expression and purification

BL21(DE3) cells (Thermo Fisher Scientific) were transformed with fIDP_pET-28a(+)-TEV plasmids according to manufacturer protocol and grown in lysogeny broth medium with 50 µg ml$^{-1}$ kanamycin. Cultures were incubated at 37 °C while shaking at 225 r.p.m. until optical density 600 of 0.6 was reached (approximately 3 h), then induced with 1 mM isopropyl β-D-1-thiogalactopyranoside and incubated for 20 h at 16 °C while shaking at 225 r.p.m. Cells were collected by centrifugation for 15 min at 3,000$g$, the supernatant was discarded and the cells were lysed in lysis buffer (50 mM NaH$_2$PO$_4$, pH 8 and 0.5 M NaCl) using a QSonica Q700 Sonicator (QSonica). Lysate was centrifuged for 1 h at 20,000$g$ and the supernatant collected and flowed through a column packed with Ni-NTA beads (Qiagen). The FRET construct was eluted with 50 mM NaH$_2$PO$_4$, pH 8, 0.5 M NaCl and 250 mM imidazole, and further purified using size-exclusion chromatography on a Superdex 200 PG column (GE Healthcare) in an ÄKTA go protein purification system (GE Healthcare). The purified FRET constructs were divided into 200-µl aliquots, flash-frozen in liquid nitrogen and stored at −80 °C in 20 mM sodium phosphate buffer, pH 7.4, with the addition of 100 mM NaCl. Protein concentration was measured after thawing and before use using ultraviolet–visible (UV–vis) absorbance at 506 nm (the peak absorbance wavelength of mNeonGreen), and purity was assessed by sodium dodecyl-sulfate polyacrylamide gel electrophoresis after thawing and before use. To verify the brightness of the FPs, we measured the UV–vis absorbance of both donor (peak absorbance wavelength of 434 nm) and acceptor molecules before each FRET assay. We used only samples that displayed an absorbance ratio Abs$_{506}$/Abs$_{434}$ of 2.8 ± 0.2, a reasonable ratio given the difference in the molar extinction coefficients of mTurquoise2 and mNeonGreen (34,000 l mol cm$^{-2}$ versus 116,000 l mol cm$^{-2}$).

### Preparation of solutions for solution space scanning

Sarcosine, PEG400, PEG2000 (Alfa Aesar), Ficoll (GE Healthcare), guanidine hydrochloride (Thermo Fisher Scientific), ethylene glycol, glycine, potassium chloride, sodium chloride, sucrose and urea (Fisher Scientific) were used without further purification. Stock solutions were made by mixing the solute with 20 mM sodium phosphate buffer, pH 7.4, with the addition of 100 mM NaCl except for experiments where the concentration of NaCl or KCl was varied, which began free of additional salt. The same buffer was used for all dilutions.

### In vitro FRET experiments

In vitro FRET experiments were conducted in black plastic 96-well plates (Nunc) with clear bottom using a CLARIOstar plate reader (BMG LABTECH). Buffer, stock solution and purified protein solution were mixed in each well to reach a volume of 150 µl containing the desired concentrations of the solute and the FRET construct, with a final concentration of 1 µM protein. Fluorescence measurements were taken from the top of the plate, at a focal height of 5.7 mm, with gain fixed at 1,020 for all samples. For each FRET construct, two repeats from different expressions with 6 or 12 technical replicates were performed in neat buffer, and two repeats from different expressions were done in every other solution condition. Fluorescence spectra were obtained for each FRET construct in each solution condition by exciting the sample in a 16-nm band centered at $\lambda = 420$ nm, with a dichroic at $\lambda = 436.5$ nm, and measuring fluorescence emission from $\lambda = 450$ to 600 nm, averaging over a 10-nm window moved at intervals of 0.5 nm. Base donor and acceptor spectra for each solution condition were obtained using the same excitation and emission parameters on solutions containing 1 µM mTurquoise2 or mNeonGreen alone[46,47].

### Calculation of FRET efficiencies and end-to-end distances

The apparent FRET efficiency ($E_f^{app}$) of each FRET construct in each solution condition was calculated by linear regression of the fluorescence spectrum of the FRET construct with the spectra of the separate donor and acceptor emission spectra in the same solution conditions (to correct for solute-dependent effects on fluorophore emission). $E_f^{app}$ was calculated using the following equation[48]:

$$E_f^{app} = 1 - \frac{F_d}{\frac{Q_a f_d}{Q_a f_a} F_s + F_d}$$

where $F_d$ is the decoupled donor contribution, $F_s$ is the decoupled acceptor contribution, $f_d$ is the area-normalized donor spectrum, $f_a$ is the area-normalized acceptor spectrum, $Q_d$ of 0.93 is the quantum yield of mTurquoise2 and $Q_a$ of 0.8 is the quantum yield of mNeonGreen[46,49]. The data for each series of solution conditions consisting of increasing concentrations of a single solute were processed as described previously[3].

### SEC and SAXS

SAXS experiments were performed at BioCAT (beamline 18ID at the Advanced Photon Source). The experiments were performed with in-line size-exclusion chromatography (SEC–SAXS). Experiments were conducted at 20 °C in 20 mM sodium phosphate, pH 7.4, with 100 mM NaCl. A total of 300 µl of samples at concentrations of approximately 4 mg ml$^{-1}$ were loaded onto a Superdex 200 Increase 10/300 column (GE Life Sciences) and run at 0.6 ml min$^{-1}$ using an ÄKTA Pure FPLC system (Cytiva). Eluent passed through a UV monitor and proceeded through the SAXS flow cell, which consists of a 1.5-mm inner diameter quartz capillary with 10-µm walls. The column to X-ray beam dead volume was approximately 0.1 ml. Scattering intensity was recorded using a Pilatus3 1 M (Dectris) detector placed 3.5 m from the sample providing access to a $q$ range from 0.003 Å to 0.35 Å. Exposures of 0.5 s were acquired every 2 s during the elution. Data were reduced at the beamline using BioXTAS RAW version 2.1.1 (refs. 50,51). The contribution of the buffer to the X-ray scattering curve was determined by averaging frames from the SEC eluent. Frames were selected as close to the protein elution as possible and, ideally, frames pre- and post-elution were averaged. When multiple peaks were observed (GS48, WT PUMA, E1A and FUS) they were deconvolved using evolving factor analysis (Supplementary Fig. 18)[52,53] and the peak with calculated molecular weight corresponding to the monomer was chosen for analysis. Final scattering profiles were generated by subtracting the average buffer trace from all elution frames and averaging curves from elution volumes close to the maximum integrated scattering intensity; these frames were statistically similar in both small and large angles. Buffer subtraction and subsequent Guinier fits (Supplementary Fig. 3), as well as Kratky transformations (Supplementary Fig. 4), deconvolution of peaks using evolving factor analysis, and molecular weight calculations based on volume of correlation[54] were done in BioXTAS RAW. Radii of gyration ($R_g$) were calculated from the slope of the fitted line of the Guinier plot at maximum $q \times R_g = 1$ using the following equation[55]:

$$\ln[I(q)] = \ln[I(0)] - \left(\frac{R_g^2}{3}\right)q^2$$

## Mammalian cell culture

HEK293T and U-2 OS cells were cultured in Corning treated flasks with DMEM (Advanced DMEM:F12) (Gibco) supplemented with 10% fetal bovine serum (Gibco) and 1% penicillin–streptomycin (Gibco). For live cell microscopy experiments, 5,000 HEK293T cells or 10,000 U-2 OS cells were plated in a μ-Plate 96-well black-treated imaging plate (Ibidi) and allowed to adhere overnight (~16 h) before transfection. Cells were incubated at 37 °C and 5% $CO_2$. Before transfection, the media was switched out with new warmed DMEM. XtremeGene HP (Sigma) was used to transfect FRET construct plasmids into HEK293T or U-2 OS cells per manufacturer's protocol. Cells were incubated at 37 °C and 5% $CO_2$ for 48 h post-transfection. NaCl stock solution was prepared by dissolving NaCl (Fisher Scientific) in 1× phosphate-buffered saline (PBS) (Gibco) and filtering using 0.2-μm filters. The solutions used for perturbations were obtained by diluting the imaging medium (1× PBS) with autoclaved deionized water to achieve hypoosmotic (100 mOsm final osmotic pressure) conditions or by adding NaCl stock solution for hyperosmotic (750 mOsm final osmotic pressure) conditions. Isosmotic (300 mOsm) conditions were obtained by adding 1× PBS. To prepare for imaging, cells were rinsed once with 1× PBS and left in 200 μl PBS (300 mOsm) just before imaging.

## Live cell microscopy

Imaging was done on a Zeiss epifluorescent microscope using a 10× 0.3 numerical aperture dry objective for whole-cell experiments or a 40× 0.9 numerical aperture dry objective for localization experiments. Excitation was done with a Colibri LED light engine (Zeiss), and data were collected on a duocam setup with two linked Flash v3 sCMOS cameras (Hamamatsu). The cells were imaged in an ambient temperature of 21 °C before and after perturbation with 150-ms exposure times. Imaging was done by exciting mTurquoise2 at 430 nm (donor and acceptor channels; Fig. 1e) or mNeonGreen at 511 nm (direct acceptor channel; Fig. 1e). Emitted light was passed on to the camera using a triple bandpass dichroic (467/24, 555/25, 687/145). When measuring FRET, emitted light was split into two channels using a downstream beamsplitter with a 520-nm cutoff. For each perturbation, the cells were focused using the acceptor channel and imaged before manually adding water (hypoosmotic condition), PBS (isosmotic condition) or NaCl solution (hyperosmotic condition) and pipetting up and down ten times to ensure mixing. Imaging was typically completed in ~45 s.

## Image analysis

Images were analyzed using ImageJ[56]. Images collected before and after osmotic challenge, containing three channels each, were stacked and aligned using the StackReg plugin with rigid transformation (Supplementary Fig. 19)[57]. The aligned image was segmented on the basis of the donor channel before perturbation. Segmentation was done using a fixed threshold that selected only pixels with intensities between 1,500 and 40,000. The resulting mask was processed using the Open and Watershed binary algorithms of ImageJ. Cells were selected using the Analyze Particles option of ImageJ, picking only those with an area between 65 μm² and 845 μm² and with a circularity of 0.1–1.0. The resulting regions of interest were averaged in each channel at each time point. The resulting cells were filtered to remove cells with an intensity over 10,000 (to correlate with in vitro experiment concentrations, see Supplementary Fig. 20) and cells where the absolute change in direct acceptor emission was over 2,000 (which tended to be cells that moved or lifted off the coverslip during measurement). To correct for donor bleedthrough and cross-excitation, cells were transfected with the mTurquoise2 or mNeonGreen construct only, the cells were imaged and analyzed using the same protocol as previously mentioned, and correlation plots were generated to determine percent bleedthrough and cross-excitation (Supplementary Fig. 21). The final filtering step removed cells with a corrected donor/acceptor ratio that was negative or higher than 6. Cell FRET efficiency before and after perturbation

($E_{f,before}^{cell}$ and $E_{f,after}^{cell}$, respectively) was calculated by $E_f^{cell} = \frac{F_A}{F_D + F_A}$. The resulting dataset is available as Supplementary Data 3. The number of cells measured for each construct and condition from this dataset are summarized in Supplementary Data 4. Analysis code is available as an ImageJ macro[58].

Images for localization experiments contained three channels that were stacked and aligned using the StackReg plugin with rigid transformation. The multipoint tool was used to manually select one 10-μm² circle in the cytoplasm and a second in the nucleus for each cell. The resulting measurements were filtered to remove cells with an intensity over 10,000 (to correlate with in vitro experiment concentrations). Cell FRET efficiency was calculated as previously stated. The resulting dataset is available in Source Data Fig. 4.

## Concentration dependence of microscopy experiments

Protein aliquot samples were diluted into a series of varying concentrations using 20 mM sodium phosphate, 100 mM NaCl, pH 7.4 buffer. Samples were prepared on a μ-Plate 96-well black-treated imaging plate (Ibidi). Fluorescent beads (Phosphorex) were added to the prepared aliquots to ensure focus on the bottom of the well. Imaging parameters were the same parameters as were used for the live cell microscopy experiments. For analysis, the center of the images were selected and the average pixel intensities were measured. To correlate emission with concentration, we plotted protein concentration against direct acceptor emission (Supplementary Fig. 20).

## Label-free peptide synthesis and purification

WT PUMA and shuffled sequences were prepared via standard microwave-assisted solid-phase peptide synthesis protocols using a Liberty Blue automated microwave peptide synthesizer (CEM) and ProTide Rink Amide resin (CEM). Fluorenylmethoxycarbonyl deprotection was achieved by treatment with 4-methylpiperidine (20% v/v) in dimethylformamide (Sigma-Aldrich), and fluorenylmethoxycarbonyl amino acids were activated using N,N'-diisopropylcarbodiimide (Sigma-Aldrich) and Oxyma Pure (CEM). Peptides were N-terminally acetylated and C-terminally amidated. After synthesis, the peptidyl resins were filtered and rinsed with acetone and air-dried. The crude peptides were cleaved from the resin for 4 h at room temperature with a 92.5% trifluoroacetic acid, 2.5% $H_2O$, 2.5% 3,6-dioxa1,8-octane-dithiol, 2.5% triisopropylsilane cleavage solution, precipitated with cold diethyl ether and centrifuged at 4,000 r.p.m. for 10 min at 4 °C. After centrifugation, the supernatants were discarded and the pellets were dried under vacuum overnight. Crude peptides were purified by high-performance liquid chromatography using an Agilent 1260 Infinity II HPLC instrument equipped with a preparative scale Phenomenex Kinetex XB-C18 column (250 mm × 30 mm, 5 μm, 100 Å) (Supplementary Fig. 22). Peptides were eluted with a linear gradient of acetonitrile–water with 0.1% trifluoroacetic acid. The target fractions were collected, rotovapped and lyophilized. Purified peptides were analyzed by mass spectrometry using a Q Exactive Hybrid Quadrupole-Orbitrap mass spectrometer (Thermo Fisher Scientific) (Supplementary Fig. 23 and Supplementary Table 5).

## CD spectroscopy

Lyophilized protein constructs were weighed and dissolved in a 20 mM sodium phosphate, 100 mM NaCl buffer at pH 7.4 to make a 200 μM stock. The stock was diluted into a concentration series to measure the CD spectra. CD spectra were measured using a JASCO J-1500 CD spectrometer with a 1 cm quartz cell for 1 μM and 2 μM protein concentration and 0.1 cm quartz cell for other concentrations (Starna Cells) using a 0.1-nm step size, a bandwidth of 1 nm and a scan speed of 200 nm min⁻¹ between 260 nm and 190 nm. Each spectrum was measured seven times and averaged to increase the signal-to-noise ratio. The buffer control spectrum was subtracted from each protein spectrum. CD spectra were normalized using UV 280 nm absorbance

to eliminate the small concentration difference between different protein constructs.

## All-atom simulations of constructs with FPs

All-atom simulations were performed of full-length FRET constructs consisting of mTurquoise2 and mNeonGreen surrounding an intrinsically disordered region (IDR). FP models were constructed from program database files 4AR7 (mTurquoise2)[59] and 5LTR (mNeonGreen)[60]. Simulations were performed using the ABSINTH implicit solvent model and CAMPARI Monte Carlo simulation engine[61].

All excluded volume interactions were present (that is, the repulsive component of the Lennard–Jones potential was turned on), while the attractive component of the Lennard–Jones potential was only turned on for residues within the IDR and limited only to intra-IDR interactions by varying the inherent Lennard–Jones parameters of all atoms outside of the IDR. Beyond these two components, all additional non-bonded Hamiltonian terms (that is, long- and short-range electrostatics and solvation effects) were turned off.

For the GS0 construct, the only backbone degrees of freedom available were associated with the set of flexible residues that connect the two beta barrels. From thousands of short independent simulations we subselected an ensemble of 1,000 distinct conformations that, on average, reproduced the experimentally measured SAXS scattering data for the GS0 construct (Supplementary Fig. 5a). This GS0 ensemble was then used to define the starting configurations of mTurquoise2, mNeonGreen and other non-GS components of the constructs for all other GS simulations.

For each of the other GS repeat lengths (8, 16, 24, 32 and 48), we performed simulations in which the attractive Lennard–Jones potential was scaled from 0.30 (random coil) to 0.62 (compact globule) in steps of 0.02. For each combination of GS length and Lennard–Jones strength, we performed 1,000 independent simulations (that is, 85,000 independent simulations in total). Each simulation was run in a spherical droplet with a radius of 500 Å for 100,000 Monte Carlo steps. The first 50,000 steps were discarded as equilibration, and conformations were then sampled every 5,000 steps. As such, each independent simulation generated 10 conformations, such that each GS/Lennard–Jones combination generated a 10,000-conformer ensemble. Having performed this set of simulations, we calculated predicted scattering profiles for each independent simulation using FoXS software, as described previously[62,63]. To assess the agreement between each short simulation and the experimental scattering data we computed $\chi^2_{\text{free}}$, a parameter explicitly developed to assess the goodness-of-fit for scattering data[54]. We generated subensembles with scattering curves that quantitatively reproduced the experimental data at each of the GS repeat lengths (Supplementary Fig. 5a).

Finally, using the SAXS-matched subensembles, we computed the distance between the centers of the two FP beta barrels (Supplementary Fig. 5b). The resulting inter-beta barrel distances are in excellent agreement with distances measured from ensemble FRET experiments. For Fig. 2b, these end-to-end distances ($R_e$) were converted to simulated FRET efficiency using $E_f = R_0^6/(R_0^6 + R_e^6)$, assuming $R_0$, the Förster distance for the mTurquoise2–mNeonGreen FRET pair, to be 62 Å (ref. 46). The final subensembles for each GS repeat length and the associated data are provided[58]. Simulation analysis was performed with SOURSOP (https://soursop.readthedocs.io/).

## All-atom simulation of IDP-only and sequence feature analysis

Simulations of label-free IDP sequences shown in Supplementary Fig. 10 were done using the CAMPARI simulation suite and the ABSINTH forcefield[61,64]. For each sequence, five independent simulations were run at 310 K using $8 \times 10^7$ Monte Carlo steps (following $1 \times 10^7$ steps of equilibration) starting from random conformations to ensure proper sampling. Protein conformations were written out every 12,500 steps. The end-to-end distance and the helicity of the simulated conformation

ensembles were determined using the MDTraj Python library. Sequence features shown in Fig. 3a,b and Supplementary Fig. 13 were evaluated using localCIDER.

## Reporting summary

Further information on research design is available in the Nature Portfolio Reporting Summary linked to this article.

## Data availability

All data needed to evaluate the conclusions in the paper are present in the paper and its Supplementary Information, as well as on the accompanying GitHub repository available at https://github.com/sukeniklab/IDP_structural_bias. All the plasmids used in this study are available from the corresponding author upon reasonable request. Some figures make use of program database structures with accession codes 4AR7 and 5LTR. Source data are provided with this paper.

## Code availability

All code used to produce the analysis and figures in this paper are available at the accompanying GitHub repository: https://github.com/sukeniklab/IDP_structural_bias.

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

## Acknowledgements

We thank J. A. Caro, M. Gebala, A. LiWang, J. Riback, P. S. Romero-Pérez, H. B. Schmidt, M. Thompson and FNZ for helpful comments and discussion. We are indebted to J. Hopkins, S. Chakravarthy and all BioCAT beamline staff at the Advanced Photon Source at Argonne National Laboratory, I. Rajkovic and all BioSAXS beamline staff at SLAC National Accelerator Laboratory, and G. Hura and all staff at the SIBYLS beamline at Lawrence Berkeley Laboratory for assistance with SAXS measurements. We thank M. Sadqi of the Center for Cellular and Biomolecular Machines (CCBM) for help with mass spectrometry. Research reported in this publication was supported by the NIH under award R35GM137926 to S.S. K.G. was supported by a fellowship from NSF-CREST Center for CCBM at UC Merced, grant no. NSF-HRD-1547848. This research used the Advanced Photon Source at Argonne National Laboratory under contract no. DE-AC02-06CH11357, proposal no. 75514. We acknowledge computing time on the MERCED cluster at UC Merced, NSF grant ACI-1429783, and on the XSEDE computational infrastructure framework, grant no. TG-MCB190103 to A.S.H. and S.S., supported by NSF grant ACI-154856.

## Author contributions

S.S. conceptualized and led the project. D.M. designed and performed all in vitro experiments and analysis with the help of R.M. K.G. designed and performed all live cell experiments and analysis with the help of N.M.S. and G.K. E.F., assisted by E.C.-Z., performed in vitro controls. A.R.P. synthesized, purified and characterized unlabeled PUMA peptides with assistance from A.D.M. E.W.M. assisted D.M. with SAXS analysis. A.S.H. and F.Y. designed, ran and analyzed simulations. S.S., D.M., K.G. and A.S.H. wrote and revised the paper.

## Competing interests

The authors declare no competing interests.

## Additional information

**Correspondence and requests for materials** should be addressed to Shahar Sukenik.

# Reporting Summary

## Statistics

For all statistical analyses, confirm that the following items are present in the figure legend, table legend, main text, or Methods section.

| n/a | Confirmed | |
|---|---|---|
| ☐ | ☒ | The exact sample size (*n*) for each experimental group/condition, given as a discrete number and unit of measurement |
| ☐ | ☒ | A statement on whether measurements were taken from distinct samples or whether the same sample was measured repeatedly |
| ☐ | ☒ | The statistical test(s) used AND whether they are one- or two-sided *Only common tests should be described solely by name; describe more complex techniques in the Methods section.* |
| ☒ | ☐ | A description of all covariates tested |
| ☒ | ☐ | A description of any assumptions or corrections, such as tests of normality and adjustment for multiple comparisons |
| ☐ | ☒ | A full description of the statistical parameters including central tendency (e.g. means) or other basic estimates (e.g. regression coefficient) AND variation (e.g. standard deviation) or associated estimates of uncertainty (e.g. confidence intervals) |
| ☐ | ☒ | For null hypothesis testing, the test statistic (e.g. *F*, *t*, *r*) with confidence intervals, effect sizes, degrees of freedom and *P* value noted *Give P values as exact values whenever suitable.* |
| ☒ | ☐ | For Bayesian analysis, information on the choice of priors and Markov chain Monte Carlo settings |
| ☒ | ☐ | For hierarchical and complex designs, identification of the appropriate level for tests and full reporting of outcomes |
| ☒ | ☐ | Estimates of effect sizes (e.g. Cohen's *d*, Pearson's *r*), indicating how they were calculated |

*Our web collection on statistics for biologists contains articles on many of the points above.*

## Software and code

Policy information about availability of computer code

| Data collection | in vitro FRET data collected using BMG MARS software v5.4R3. SEC and SAXS data collected on the APS beamline using in-house software. CD data collected using JASCO Spectra Manager v2.14.06. Imaging data collected using Zen Blue v2.6. All simulations performed using CAMPARI v2 |
|---|---|
| Data analysis | Unless otherwise stated, all data was analyzed using custom python code available at https://github.com/sukeniklab/IDP_structural_bias. SAXS data analyzed using BioXTAS RAW v2.1. Microscopy images analyzed with ImageJ v1.53. Simulation analysis performed using MDTraj v1.9.4. |

For manuscripts utilizing custom algorithms or software that are central to the research but not yet described in published literature, software must be made available to editors and reviewers. We strongly encourage code deposition in a community repository (e.g. GitHub). See the Nature Portfolio guidelines for submitting code & software for further information.

## Data

Policy information about availability of data

All manuscripts must include a data availability statement. This statement should provide the following information, where applicable:
- Accession codes, unique identifiers, or web links for publicly available datasets
- A description of any restrictions on data availability
- For clinical datasets or third party data, please ensure that the statement adheres to our policy

All data needed to evaluate the conclusions in the paper are present in the paper and its supporting information, as well as on the accompanying github repository available at https://github.com/sukeniklab/IDP_structural_bias. All the plasmids used in this study are available from the corresponding authors upon reasonable request. Some figures make use of PDB structures with accession codes 4AR7 and 5LTR. Source data are provided with this paper.

## Research involving human participants, their data, or biological material

Policy information about studies with human participants or human data. See also policy information about sex, gender (identity/presentation), and sexual orientation and race, ethnicity and racism.

| | |
|---|---|
| Reporting on sex and gender | N/A |
| Reporting on race, ethnicity, or other socially relevant groupings | N/A |
| Population characteristics | N/A |
| Recruitment | N/A |
| Ethics oversight | N/A |

Note that full information on the approval of the study protocol must also be provided in the manuscript.

# Field-specific reporting

Please select the one below that is the best fit for your research. If you are not sure, read the appropriate sections before making your selection.

☒ Life sciences　　　　☐ Behavioural & social sciences　　　　☐ Ecological, evolutionary & environmental sciences

For a reference copy of the document with all sections, see nature.com/documents/nr-reporting-summary-flat.pdf

# Life sciences study design

All studies must disclose on these points even when the disclosure is negative.

| | |
|---|---|
| Sample size | Sample sizes are provided for all experiments where N>30 in Table S4. Otherwise, individual datapoints are shown on the plots. Sample sizes are obtained from the numbers of cells imaged, with some exclusions as detailed below and in the methods. |
| Data exclusions | Some cells for live-cell data are filtered out based on specific criteria - under or over-expressed cells, cells with abnormal morphology following segmentation, and cells that have lifted off the coverslip following osmotic perturbations. This constituted < 10% of the imaged cells. |
| Replication | All experiments were conducted in two repeats at least. Repeats were conducted on different days, from different stock samples, and often from different protein expression batches. Live cell data was conducted on different passages, in different imaging chambers, and with individual transfections. All attempts at replication were successful. |
| Randomization | Different constructs were plated and imaged in different orders and different positions on well plates to ensure time prior to measurement is not a factor affecting the results |
| Blinding | Analysis of all raw data is fully automated and no blinding was used for this study |

# Reporting for specific materials, systems and methods

We require information from authors about some types of materials, experimental systems and methods used in many studies. Here, indicate whether each material, system or method listed is relevant to your study. If you are not sure if a list item applies to your research, read the appropriate section before selecting a response.

## Materials & experimental systems

| n/a | Involved in the study |
|-----|----------------------|
| ☒ | ☐ Antibodies |
| ☐ | ☒ Eukaryotic cell lines |
| ☒ | ☐ Palaeontology and archaeology |
| ☒ | ☐ Animals and other organisms |
| ☒ | ☐ Clinical data |
| ☒ | ☐ Dual use research of concern |
| ☒ | ☐ Plants |

## Methods

| n/a | Involved in the study |
|-----|----------------------|
| ☒ | ☐ ChIP-seq |
| ☒ | ☐ Flow cytometry |
| ☒ | ☐ MRI-based neuroimaging |

## Eukaryotic cell lines

Policy information about cell lines and Sex and Gender in Research

| | |
|---|---|
| Cell line source(s) | ATCC |
| Authentication | Cells were obtained directly from ATCC and no further authentication was performed |
| Mycoplasma contamination | Not tested for mycoplasma contamination |
| Commonly misidentified lines (See ICLAC register) | No commonly misidentified cells were used in this study |

