## [Peer Review File · Nature Structural & Molecular Biology]

Peer Review Information

Manuscript Title: Structural biases in disordered proteins are prevalent in the cell

Corresponding author name(s): Shahar Sukenik

Reviewer Comments & Decisions:

Decision Letter, initial version:
--

Message: 6th Oct 2022

Age:

Dear Dr. Sukenik,

Thank you for submitting your manuscript "Structural biases in disordered proteins are prevalent in the cell". Once again, please accept my sincere apologies for the unusual delay in processing your manuscript, which resulted from difficulties in obtaining referees' reports, together with our editorial team being short-staffed at the moment. Nevertheless, we now have comments from the 2 reviewers who have evaluated your manuscript below. Unfortunately, after carefully considering their comments, we cannot offer to publish your manuscript in Nature Structural & Molecular Biology.

You will see that while the referees find the work interesting and innovative, they raise concerns about the lack of a demonstrated functional relevance for the observed structural biases in IDPs, and also about the generality and conclusivity of the results based on the set of fluorophores and in vitro conditions tested.

Although we regret that we cannot offer publication at NSMB, I have discussed your manuscript and the reviewer reports with our colleagues at Nature Communications, and they would be happy to send it back for formal peer review if you transfer the manuscript there after revision. Particularly, please make sure to:

- 1- Provide further evidence of the functional relevance pertaining to the biophysical findings.
- 2- Validate the experimental work as highlighted by reviewer 1 and reviewer 2, addressing all technical concerns.

Should you wish to have your paper reviewed at Nature Communications, please use the link to the Springer Nature manuscript transfer service in the footnote. It is not necessary to reformat your paper at this point.

Your handling editor at Nature Communications would be Dr. Engi Hassaan

(engi.hassaan@nature.com). If there is anything you would like to discuss before transferring the paper, please don't hesitate to contact her by e-mail.

I am sorry we could not be more positive on this occasion. I hope that you find the referees' comments useful in deciding how best to proceed.

Sincerely,
Sara

Sara Osman, Ph.D.
Associate Editor
Nature Structural & Molecular Biology

Referee expertise:

Referee #1: IDPs, computation

Referee #2: IDPs, biophysics

Reviewers' Comments:

Reviewer #1 (Remarks to the Author):

David Moses et. al. investigate the impact of sequence variation (length and composition) in IDRs on structural compactness in vitro and in cells. In order to do so, they use FRET, SEC and SAXS. Moreover, they probe the impact of solution changes and changes in the cellular milieu on the structural properties of several IDRs. Their most important finding is that the structural biases observed in vitro are conserved in cells.

This is a very interesting and innovative study. This said, several aspects of this work could be improved.

1. The conservation of structural biases of IDRs in cells and the response to solution/milieu changes are important finding. However, the direct functional relevance of the IDR behaviour is not clear. It would be insightful to link structural biases and/or the response to milieu changes to the function of a specific IDR/IDP. Such direct link between the ability to respond to milieu changes and the function of an IDR would strengthen the impact of this work significantly.
2. Structural biases in the cell are likely affected by many factors (the authors mention PTMs and PPIs) What about cellular localization? Are all IDRs tested in the cytoplasm or the nucleus? How does the localization affect the structural properties? It is well known that the nucleus has very different milieu conditions than the cytoplasm, which will affect the measures properties.
3. One key uncertainty in this study is the impact of the FPs on IDR behaviour. It is possible that some of the tested IDR sequences interact with the FPs, which will affect compactness. It would be great if the authors could confirm some of their key findings with an alternative FP pair.

4. I am missing statistical assessments (p values) of the differences in structural biases throughout the manuscript.
5. The comparison with simulation results on page 4 seems a bit circular. Ensembles were selected to match the SAXS data. No wonder the R_g and E_f trends agree. What about the ensembles as they are produced directly by the simulations (not matched)? How do they compare?
6. How were the PUMA sequence scrambles generated? Based on what criteria?
7. Page 11: It is stated that E1A is more compact in vitro but more expanded in the cell (E_f is compared) I do not see that difference in figures 4 A and D.

Reviewer #2 (Remarks to the Author):

The study entitled "Structural biases in disordered proteins are prevalent in the cell" by Moses et al. investigates structural biases in disordered proteins in vitro and in cells. The authors investigate different IDP in reference to GS repeat proteins in vitro under different conditions by different techniques including FRET, SAXS, CD-spectroscopy or all-atom simulations. This characterization allows them to interpret the data they acquire by live cell imaging. They arrive at the conclusion that structural biases of IDPs (known from different other in vitro studies) prevail in cells and propose that the responsiveness to (changing) physicochemical properties of the cellular environment is linked to biological function in health and disease conditions. I think this is very interesting work as most of our knowledge about IDPs is derived from in vitro experiments and this study extrapolates this knowledge to cellular conditions. The paper is well written and amenable to a broad audience. However, I have some concerns that require a major revision of the manuscript:

In their earlier studies "Intrinsically disordered protein biosensor tracks the physical-chemical effects of osmotic stress on cells" by Cuevas-Velazquez et al. in Nat Commun, 2021 the authors use similar techniques like osmotic perturbations, scrambling sequences and solvent modulation constructing a biosensor from a naturally occurring LEA-IDP. Its applicability was demonstrated in several types of cells, leading to novel biological insight and knowledge such as the size-effect of vacuoles as a water resort on the individual cell level. The authors cite this work, revealing that they are well-familiar with the techniques, but I am wondering why they are not comparing the LEA-IDP effects to the IDP and GS repeat in this study.

I am puzzled by the choice of in vitro conditions the authors use. They specifically discuss on macromolecular crowding effects, why did they not compare macromolecular crowders to molecular crowders? E.g. different PEG length, Ficoll vs sucrose? What is the motivation of using the amino acids and choice of salts?

The authors openly discuss the limitations and drawbacks of their approach, which is good. I miss information why they decided on these specific fluorophores. Did they check if the fluorophores mature correctly in cells?

Compared to the IDPs, I cannot find any CD experiments for the GS repeats.

Page 13: Elongated structures under high crowding can be indeed explained. Another explanation could be the binding of molecular chaperones.

The authors use solvent perturbation as a method to study the structural biases of IDPs in cells. Temperature is another variable with high physiological relevance for IDPs, e.g. formation of stress granules under heat stress. In particular in the cell, temperature can be easily (and rapidly, e.g. by temperature jump) modulated. Have the authors conducted experiments under different incubation temperatures in cells?

Minor: Page 4: "Folded proteins are often compared to crystal structure". I guess the authors want to say something like "The structure of folded proteins is often assumed to be identical to crystal structure ...?"

****Nature Communications** is the Nature Portfolio flagship Open Access journal. If you would like this work to be considered for publication there, you can easily transfer the manuscript by following the instructions below. It is not necessary to reformat your paper. Once all files are received, the editors at Nature Communications will assess your manuscript's suitability for potential publication; they aim to provide feedback quickly, with a median decision time of 8 days for first editorial decisions on suitability. If your paper has been peer reviewed at this journal, the referee reports will also be transferred and assessed by the editorial team. In some cases, papers are accepted without further peer review, providing a rapid path to publication. The journal is also proud to offer double blind and transparent peer review options. For 2019, the 2-year impact factor for Nature Communications is 12.121 and the 2-year median is 8 (for further information on journal metrics, please visit our Nature journals metrics page). Our open access pages contain information about article processing charges, open access funding, and advice and support from Springer Nature.

****I suggest that you consider Nature Communications as a suitable venue for your work. To transfer your manuscript there, please use our manuscript transfer portal. You will not have to re-supply manuscript metadata and files, unless you wish to make modifications, but please note that this link can only be used once and remains active until used. For more information, please see our manuscript transfer FAQ page.**

Note that any decision to opt in to In Review at the original journal is not sent to the receiving journal on transfer. You can opt in to In Review at receiving journals that support this service by choosing to modify your manuscript on transfer. In Review is available for primary research manuscript types only.

** For Springer Nature Limited general information and news for authors, see
<http://npg.nature.com/authors>.

Sara Osman, Ph.D.
Associate Editor
Nature Structural & Molecular Biology

Author Rebuttal to Initial comments

Reviewer #1 (Remarks to the Author):

David Moses et. al. investigate the impact of sequence variation (length and composition) in IDRs on structural compactness *in vitro* and in cells. In order to do so, they use FRET, SEC and SAXS. Moreover, they probe the impact of solution changes and changes in the cellular milieu on the structural properties of several IDRs. Their most important finding is that the structural biases observed *in vitro* are conserved in cells.

This is a very interesting and innovative study. This said, several aspects of this work could be improved.

We thank the reviewer for their supportive comments, and for highlighting the importance of establishing that structural biases measured *in vitro* are retained in cells to the field. We also thank the reviewer for their questions and suggestions, which we address below. We have done our best to include figures inline with the response in this document. These are labeled as **Fig. RX** - but all of them are also included either in the main text (labeled **Fig. X**) or in the SI (labeled **Fig. SX**).

1. The conservation of structural biases of IDRs in cells and the response to solution/milieu changes are important finding. However, the direct functional relevance of the IDR behaviour is not clear. It would be insightful to link structural biases and/or the response to milieu changes to the function of a specific IDR/IDP. Such direct link between the ability to respond to milieu changes and the function of an IDR would strengthen the impact of this work significantly.

This question has been our main consideration when deciding which IDPs to characterize in this study. We have not properly articulated this in the previous version. The manuscript now includes a new paragraph in the discussion detailing the link between ensemble and function for all IDPs used in this study:

“The importance of IDP ensembles for molecular function has been shown or proposed for all of the naturally occurring IDPs characterized in this study. The structural preferences of the PUMA BH3 ensemble have been shown to affect its binding kinetics to MCL1 - a key event in the function of PUMA as a modulator of p53, and it has further been shown that this structural change can come about by changing the composition of the solution¹⁰. Changing the structural preferences of the p53 N-terminal ensemble affects its binding affinity to MDM2, a potent inhibitor of p53's protective function, altering downstream p53 function¹¹. FUS low-complexity region can undergo phase separation *in vitro* and *in vivo*. Recent work has shown that for monomeric low-complexity sequences, chain dimensions dominated by intramolecular interactions can quantitatively inform on intermolecular interactions in the context of phase transitions⁷⁸. The Ash1 ensemble has been shown to remain largely unperturbed by phosphorylation, indicating the need for robust activity of this key transcription factor in yeast⁶¹. Finally, a region proximal to the E1A sequence used here has been shown to be highly conserved in terms of the average end-to-end distance of its ensemble, and this length critical to its function, implicating strong selection for ensemble properties across the whole protein²⁰. ”

Importantly, in all of these cited works, as well as most other works in the field, **linking ensemble structure to function is done *in vitro*. In this work we show that the interactions that hold ensemble structural preferences together *in vitro* are the same in the cell.** We believe this finding is key to establishing the link between ensemble structure and function in the cell.

2. Structural biases in the cell are likely affected by many factors (the authors mention PTMs and PPIs) What about cellular localization? Are all IDRs tested in the cytoplasm or the nucleus? How does the localization affect the structural properties? It is well known that the nucleus has very different milieu conditions than the cytoplasm, which will affect the measured properties.

This is an excellent question. Previously, our dataset was obtained at low magnification and delineating between specific subcellular loci was difficult to perform accurately. We have repeated our in-cell measurements of all constructs, but this time using a higher magnification objective (40x, 0.9 NA) and U-2 OS cells that have a significantly larger cytoplasm than HEK293Ts.

To enable comparisons, we first measured our GS repeat sequences in both cytoplasm and nucleus in the new cell line, and compared them to the values in HEKs. We find that overall the value of E_f^{cell} is lower in U-2 OS than HEK293T. This can be explained by the different cellular composition of the embryonic kidney cells from which HEK293Ts are derived compared to the osteosarcoma cells of U-2 OS. Further studies of this line-to-line variability are ongoing in our lab. Regardless, the trend in E_f^{cell} vs. sequence length in GS repeats in both cell lines remains linear and the slope is the same within error (Now **Fig. S16**, and summarized in **Fig. R1** below), indicating that the scaling behavior of GS repeats is the same in both cell lines. Furthermore, the slope for cytoplasm and nucleus for GS repeat lengths are the same within error. This indicates that, at least for GS repeats, localization does not change the structural preferences of the ensembles.

Fig. R1. Linear fits of the medians with fit errors shown by the shaded region for HEK293T (gray), U-2 OS cytoplasm (purple), and U-2 OS nucleus (blue). See also **Fig. S16**.

We next compared E_f^{cell} of four IDR constructs in U-2 OS cells in the cytoplasm and the nucleus. To facilitate effective comparison, E_f^{cell} measurements were normalized to a GS-repeat sequence of the same length in the respective environments (**Figs. R2, 4E**). Three of the four sequences showed no significant change between the cytoplasm and the nucleus. A significant difference ($P < 0.0001$) was observed for FUS, and indicates that FUS is more expanded in the nucleus than it is in the cytoplasm. We hypothesize that this may be a result of RNA binding in the nucleus: Despite lacking RNA binding motifs, the N-terminal low complexity domain of FUS which is used here has been reported to bind to nuclear-abundant RNA⁶. This is now included in the text:

“We next wanted to see how the localization of IDPs in the cell might affect their ensembles. We reasoned that different organelles have different physical-chemical compositions, and this may affect the ensemble preferences encoded in IDP sequences⁶². To test this idea, we measured E_f^{cell} in the cytoplasm and nucleus of U-2 OS cells for all our sequences. GS repeats showed the same E_f^{cell} in both cytoplasm and nucleus within error, indicating their ensemble is unaffected by changes in localization (**Fig. S16**). All E_f^{cell} measurements were normalized to a GS repeat of the same length (**Fig. 4E**). Most sequences showed no significant difference between the cytoplasm and the nucleus. This is in line with our results thus far: if moving these sequences from aqueous buffers to the cellular environment induced little change in ensemble structure, we expect the same to happen moving from the cytoplasm to the nucleus. An exception was observed for the FUS low complexity domain which was significantly more expanded in the nucleus (**Fig. 4E**). This might be due to its ability to interact with nuclear-abundant RNA^{63,64}.”

Fig. R2. E_f^{cell} of four IDR constructs measured in the cytoplasm (C) and nucleus (N) of U-2 OS cells and normalized to the E_f^{cell} of an equivalent GS linker. Each box represents the 25th and 75th percentiles with the median shown as the black line and the whiskers showing the minimum and maximums for each construct. Each circle corresponds to a single cell. Asterisks denote the significance between distributions determined by a Mann-Whitney test (**** indicates $p < 0.0001$). See also **Fig. 4E**.

3. One key uncertainty in this study is the impact of the FPs on IDR behaviour. It is possible that some of the tested IDR sequences interact with the FPs, which will affect compactness. It would be great if the authors could confirm some of their key findings with an alternative FP pair.

Repeating the measurements with a new set of FPs would be problematic due to the specifics of our microscopy setup and the need to redo all calibration experiments with a new FP pair. Instead, we measured the constructs with the donor and acceptor location flipped (i.e., mNeonGreen on the N-terminus and mTurquoise2 on the C-terminus) (Figs. 5A, R3A). We reasoned that since each FP has a different surface area chemistry (Figs. R3B, S22A) and different amino acid sequences at its opposite termini (Figs. R3C, S22B), flipping the FPs will place the IDR in a different environment and potentially disrupt any of its interactions with the FP.

Figure R3. Original and flipped GS16 repeat constructs. **(A)** Original FRET (top) construct consisting of an IDR between two fluorescent proteins that serve as a FRET donor and a FRET acceptor and the flipped construct (bottom) with the FRET pairs on the opposite end. **(B)** Surface electrostatic analysis using APBS⁹ shows different surface charges for mNeonGreen and mTurquoise2. The N and C termini are labeled as cyan and yellow spheres, respectively. **(C)** Sequences of GS16 with the nearest 20 residues from the flanking fluorescent proteins, aligned using Clustal Omega^{10,11}. Color codes are from CIDER analysis¹². Red: negative charge; blue: positive charge; black: hydrophobic residues; green: polar residues; orange: aromatic residues. Blue and green boxes show residues at the terminals of mTurquoise2 and mNeonGreen, respectively, connected to the GS-repeat sequence. See also Figs. 5A and S22.

We first tested this idea using a GS-repeat construct. Measurements of E_f^{app} *in vitro* showed basal differences between the original and flipped constructs, with the flipped construct showing a higher FRET efficiency indicating a more compact conformation (**Figs. 5B, R4**). Since GS repeats have no charged residues, and since all GS-repeat sequences are the same at the N and C terminals, we hypothesized that the effect of flipping the FPs on E_f^{app} could come from two sources:

- 1) A change in interaction between the FPs themselves, which would likely be electrostatically driven because of the different surface orientations (**Figs. S22A, R3B**).
- 2) A change in the effective length of the linker because of partial unfolding of the N- or C-terminal of each FP.

Fig. R4. Comparison of average E_f^{app} of original and flipped GS16 constructs. Error bars are SD of the data (N=12). See also **Fig. 5B**.

To test which of these scenarios is more likely, we first measured the effect of salt concentration on both constructs (**Figs. S23A, R5A**). Our experiments show that the screening effect of the salt, as shown by the exponential decay constant of E_f^{app} vs [NaCl], is identical in the original (6.8 ± 0.7) and flipped (6.8 ± 0.6) constructs (**Figs. S23A,B, R5A,B**). This means that while there are electrostatic interactions that affect E_f^{app} , they are the same in the original and flipped constructs, ruling out option (1).

The behavior of E_f^{app} also shows that a difference between the original and flipped construct emerges at higher salt concentrations, with a larger slope for the original construct (**Figs. S23A,C, R5A,C**). An analysis of this slope for our original GS-repeat sequences of different lengths shows that as the sequence grows longer, the slope becomes larger (**Fig. S23C, R5C**). The flipped construct shows a slope that is smaller than expected for the original construct, indicating its length might be effectively shorter. This is in line with the higher E_f^{app} for this construct. We hypothesized that this might be due to a tighter packing of the C- or N-terminal of one of the FPs. Analysis of the fluorescence spectra showed that while mTurquoise2 has virtually no shift in peak position upon tethering, the peak of mNeonGreen significantly shifts depending on whether it is or how it is untethered (**Figs. S23D, R5D**). We conclude that the

changes in basal E_f^{app} in the flipped and original GS16 sequences result primarily from changes in mNeonGreen, and propose this is due to tethering rather than from any specific interactions between the GS chain and the FPs themselves. This further validates the use of these constructs as points of reference.

Figure R5. (A) Effect of NaCl titration on average E_f^{app} of GS-repeats and the flipped GS16 construct. Experimental data was fit to an exponential decay with a sloping baseline, $E_f^{app}([NaCl]) = Ae^{(-k[NaCl])} + m[NaCl] + b$. In this equation k is a decay constant that indicates the effect of screening of electrostatic interactions on ensemble structure, and m is a linear slope that accounts for the specific interactions of the ions at higher concentrations^{13–15}. (B) Comparison of k obtained from the fit of the original and flipped GS16 constructs. The identical value of k indicates that electrostatic interactions cannot explain the difference in E_f^{app} between the two constructs. (C) Slope m vs. the length of all GS-repeat sequences. All original GS repeats show a linear relationship between m and length. The flipped GS16 construct falls below this line, indicating a tighter packing of one or both of the FPs. (D) Comparison of the peak emission wavelengths for mTurquoise2 (left) and mNeonGreen (right) untethered, in the original GS16 construct, and in the flipped GS16 construct. For mNeonGreen, $P < 0.0001$ for both untethered vs. GS16 original and untethered vs. GS16 flipped using Student's t-test. See also **Fig. S23**.

Next we measured flipped versions of some of the constructs we had previously measured in live cells. For GS16, the live-cell results once again recapitulated the *in vitro* results, indicating that whatever changes occurred in the flipped construct *in vitro* also occurred inside the cell (**Figs. 5C, R6A**). For naturally occurring IDRs, the flipped constructs showed overall lower E_f^{cell} compared to the original values, indicating that the flipped constructs were more expanded than the original (**Figs. 5D, R6A**). This is in contrast to our GS16 construct, and could possibly be attributed to electrostatic interactions with charged residues in the chain. While flipped Ash1 and p53 displayed only slightly lower E_f^{cell} distributions compared to the original constructs, flipped

E1A displayed a dramatically lower E_f^{cell} , indicating that flipping the FPs significantly expanded the sequence (**Fig. 5D** and **Fig. R6A**).

Fig. R6: (A) E_f^{cell} of selected constructs measured in HEK293T cells for original and flipped constructs. (B) Response to osmotic challenge of each construct expressed as change in E_f^{cell} before and after the challenge (ΔE_f^{cell}). N > 1400 for all violin plots. See also **Fig. 5C-E**.

This large change in E_f^{cell} could be a result of strong attractive interactions between the E1A sequence and one or both of the FPs in the original sequence. We reasoned that if the ensemble changed dramatically in the flipped construct, this would also change the way the sequence would respond to cell volume perturbations. Indeed, our experiments show that while Ash1 and p53 show a similar response in the flipped and original construct, the outlying response of E1A has now flipped completely (**Figs. 5E, R6B**).

While these experiments point to an artifact in the original E1A construct that may dominate the measured ensemble, we point to the fact that, in nature, all IDRs studied here, and most of those studied in the literature, are attached to folded domains with various surface area chemistries. Here we show, for the first time, that these interactions can drive measurable changes to IDR ensembles, and also change the way they respond to changes in the intracellular environment. These experiments and discussions are now incorporated into the text under the heading: “**Interactions between IDPs and their tethered folded domains**”

4. I am missing statistical assessments (p values) of the differences in structural biases throughout the manuscript.

Our live-cell datasets are large (usually N > 1000). This means that all distributions shown as violin plots will be significantly different from each other with very small P-values, even if the changes are barely visible by eye or meaningless in terms of E_f^{cell} differences. This causes P-values to be misleading. Where relevant, we have added significance values, either as a P-value or as the overlap between the median 50% to help the reader assess differences between distributions. In cases where the datasets are smaller, we have added the P-value as

asterisks to the graphics directly. We also point out that we do not compare E_f^{app} to E_f^{cell} directly, since these observations are done in different experiments and different methods.

5. The comparison with simulation results on page 4 seems a bit circular. Ensembles were selected to match the SAXS data. No wonder the R_g and E_f trends agree. What about the ensembles as they are produced directly by the simulations (not matched)? How do they compare?

While we understand the reviewer's point, we would like to stress that the result is NOT circular - we pick a simulated ensemble that matches the raw experimental SAXS scattering curves, and then calculate E_f^{app} of that simulated ensemble through the distance between the center of mass of the FPs. Because it cannot be assumed that experimentally determined E_f^{app} obtained from FRET measurements, would necessarily match the calculated simulated E_f^{app} obtained from SAXS measurements, the fact that they do match is meaningful. There is a large body of work illustrating that this exact result is often not obtained - i.e., that one cannot necessarily predict IDR global dimensions based on end-to-end distance, when the protein in question deviates from homopolymer expectations^{16,17}. Notably, we know that radius of gyration vs. number of residues does not follow a simple homopolymer model due to the contributions of the FPs. As such, the ability to select sub-ensembles via one experimental metric and have that ensemble reproduce a second metric is a commonly-used approach to assess the validity of an ensemble. We recognize that this is a necessary but not sufficient criterion to demonstrate agreement, but for our purposes here the simulations are used as a confirmatory sanity check (as opposed to being used to drive new hypotheses). We have clarified the role of the simulations in offering a computational thought experiment (as opposed to a hard prediction) in the text:

"Finally, we conducted all-atom simulations of all GS-repeat sequences to enable a molecular benchmark between SAXS and FRET results. Our simulations assumed that the FPs only take up space (i.e., are non-interacting) and that GS repeats behave like homopolymers. From these simulations, ensembles were selected to quantitatively match the SAXS scattering data (**Fig. S5**). These ensembles reproduced the GS length-dependent E_f^{app} values as well, indicating that the simulation conditions at least managed to reproduce our experimental results (**Fig. 2B,F**). The application of one experimental dataset as a constraint to assess simulations against an orthogonal experimental dataset has been used previously to assess unfolded protein ensembles to great effect^{49,50}."

6. How were the PUMA sequence scrambles generated? Based on what criteria?

The PUMA sequences were designed to sample varying degrees of charge clustering, as measured by the parameter κ ¹². We have provided the values of κ in **Fig. 3A**, and provided further clarification in the text:

“The three scrambles of WT PUMA were designed to have varying degrees of charge clustering, as measured by the parameter κ (kappa) in CIDER⁵⁶ (sequences S1-3, **Fig. 3A,B**)”

7. Page 11: It is stated that E1A is more compact *in vitro* but more expanded in the cell (Ef is compared) I do not see that difference in figures 4 A and D.

We have removed this wording.

Reviewer #2 (Remarks to the Author):

The study entitled “Structural biases in disordered proteins are prevalent in the cell” by Moses et al. investigates structural biases in disordered proteins *in vitro* and in cells. The authors investigate different IDP in reference to GS repeat proteins *in vitro* under different conditions by different techniques including FRET, SAXS, CD-spectroscopy or all-atom simulations. This characterization allows them to interpret the data they acquire by live cell imaging. They arrive at the conclusion that structural biases of IDPs (known from different other *in vitro* studies) prevail in cells and propose that the responsiveness to (changing) physicochemical properties of the cellular environment is linked to biological function in health and disease conditions. I think this is very interesting work as most of our knowledge about IDPs is derived from *in vitro* experiments and this study extrapolates this knowledge to cellular conditions. The paper is well written and amenable to a broad audience. However, I have some concerns that require a major revision of the manuscript:

We thank the reviewer for their positive assessment of our work. We also thank the reviewer for their questions and suggestions, which are all addressed below. We have done our best to include figures inline with the response in this document. These are labeled as **Fig. RX** - but all of them are also included either in the main text (labeled **Fig. X**) or in the SI (labeled **Fig. SX**).

In their earlier studies “Intrinsically disordered protein biosensor tracks the physical-chemical effects of osmotic stress on cells” by Cuevas-Velazquez et al. in Nat Commun, 2021 the authors use similar techniques like osmotic perturbations, scrambling sequences and solvent modulation constructing a biosensor from a naturally occurring LEA-IDP. Its applicability was demonstrated in several types of cells, leading to novel biological insight and knowledge such as the size-effect of vacuoles as a water resort on the individual cell level. The authors cite this work, revealing that they are well-familiar with the techniques, but I am wondering why they are not comparing the LEA-IDP effects to the IDP and GS repeat in this study.

We appreciate the reviewer’s comment. Indeed, in the prior work mentioned (on which Sukenik is a co-corresponding author and which several of the authors of this manuscript have co-authored), we developed the SED1 sensor to detect changes in cellular crowding. However, sensing changes in the cell is not the focus of this specific work - instead, we ask if structural preferences seen *in vitro* in IDPs persist in the cellular milieu. As such, this comparison, while interesting, is outside the scope of this paper.

I am puzzled by the choice of in vitro conditions the authors use. They specifically discuss on macromolecular crowding effects, why did they not compare macromolecular crowders to molecular crowders? E.g. different PEG length, Ficoll vs sucrose? What is the motivation of using the amino acids and choice of salts?

We thank the reviewer for these suggestions. We use various amino acids, salts, denaturants, other small osmolytes, and crowders as chemical probes to compare the responses of various IDRs to different types of changes in their physical-chemical environment¹³. The solutes themselves are not intended to directly mimic the cellular environment (or any of its myriad constituents), but rather to chemically probe the ensemble structure of our constructs. Specifically we use the amino acid glycine, a naturally occurring osmolyte that has been found to stabilize folded proteins in the face of osmotic pressure²⁰⁻²². Salts probe the contribution of electrostatics to ensemble structure by screening attractive or repulsive charge interactions¹⁵. This is now explained briefly in the main text:

“The solutes we added as chemical probes in this case were salts, amino acids, polymeric crowders and their monomeric units, and denaturants. We stress that these solutes were not intended to directly mimic the cellular environment, but rather to probe the response of the ensemble to changes in solution chemistry. We measured the change in FRET efficiency $\Delta E_f^{app} = E_{f,solute}^{app} - E_{f,buffer}^{app}$ for all GS repeat lengths (**Fig. S6**). As expected, GS repeats of all lengths responded identically to each of the solution conditions we created (**Fig. S6**).”

Furthermore, following the reviewer’s suggestion, we have now included sucrose, ethylene glycol and PEG 400 in **Fig. 4F** (and **Fig. R7**). In line with our previous results¹³, we found that the monomers do not increase compaction of any of the IDPs, and that PEG 2000 shows a greater increase in ΔE_f^{app} than the smaller PEG 400 at equal monomer-molar concentrations. We now make this point in the text:

“In line with our previous results⁵, we found that PEG 2000 produces greater increases in E_f^{app} than the smaller PEG 400 at equal monomer-molar concentrations, and that the monomer units of the crowders (sucrose, ethylene glycol) produce relatively small changes in the dimensions of the IDPs.”

In general, our reluctance to include large macromolecular crowders stems from our and others’ extensive past experience with these substances. Specifically, PEG is far from inert and commonly interacts with specific protein moieties, leading to results that are difficult to interpret²³⁻²⁵. This is also apparent in our own data, showing a dramatic increase in E_f^{app} for several sequences that can also be caused by aggregation or phase separation. Indeed, high concentrations of high MW PEG were shown numerous times to induce phase separation²⁶. Our new experiments with PEG2000 illustrate this point precisely: We use up to 25% w/w of this polymer - less than the crowding observed in live cells, and see that E_f^{app} jumps very high in

some sequences (Ash1 being case in point, **Fig. S14**). This behavior simply isn't observed inside the cell.

Fig. R7. Solution space scans of constructs incorporating naturally occurring IDPs, with results expressed as ΔE_f^{app} , the difference between E_f^{app} of an IDP construct in a given solution condition and in a dilute buffer. White dots: ΔE_f^{app} of IDP. Black dashed lines: interpolated ΔE_f^{app} of a GS-repeat sequence of the same length as the IDP. Blue-green shaded regions between white dots and black dashed lines: difference between ΔE_f^{app} of IDP and GS repeats. Heatmap backgrounds: red shows more sensitivity (more expansion or compaction) than a GS-repeat sequence of the same length; blue shows less sensitivity than GS repeats; white shows the same sensitivity as GS repeats; deeper shades show greater difference in sensitivity from GS repeats. Shaded regions on left side of cells for solutes NaCl and KCl: approximate range of concentrations within which electrostatic screening is the dominant effect; the leftmost two points of each series, since they are within that range, are not used in the assignment of background color.

The authors openly discuss the limitations and drawbacks of their approach, which is good. I miss information why they decided on these specific fluorophores. Did they check if the fluorophores mature correctly in cells?

Previous studies have shown the mTurquoise2:mNeonGreen pair to be one of the best FRET pairs for in-cell work²⁷. We have previously used other FPs (such as mEGFP:mCherry²⁸ or mCitrine:mCerulean²⁹) - in these cases either the dynamic range of detection was reduced significantly (in the case of dimmer red FPs) or there were issues with dimerization (in the case of mCitrine and mCerulean). For FPs with wavelengths lower than mTurquoise2 background fluorescence becomes an issue. Thus we have selected this specific pair. This is now states in the main text:

“The monomeric fluorescent proteins mTurquoise2 and mNeonGreen have high quantum yields, fast maturation and are highly photostable^{34,83}. Compared to other FRET pairs, mTurquoise2 and mNeonGreen have a higher FRET efficiency with little cell to cell variation and were therefore selected as our FRET pair in our experiments³⁴.”

In terms of maturation, mTurquoise2 and mNeonGreen mature in less than an hour (<10 minutes for mNeonGreen³⁰, 33.5 minutes for mTurquoise2³¹). We transfect cells 24-48 hours prior to imaging, ensuring maturation is completed for the bulk of the expressed construct population. We have now added a sentence to clarify this in the Methods section:

“Cells were incubated at 37 °C and 5% CO₂ for 48 hours post transfection, ensuring that maturation of the FRET pairs (which occurs in 10-30 minutes^{94,95}) does not alter our results.”

Compared to the IDPs, I cannot find any CD experiments for the GS repeats.

CD experiments were performed only for PUMA and its variants, since we wanted to ensure that structural preferences, specifically local helical structure, within the ensemble are depleted. GS repeats have been shown experimentally to lack such structural preferences³²⁻³⁵, and our own simulations similarly show no helicity in GS repeat sequences. To explain this point more clearly, we have revised our argument for using GS-repeat sequences as a benchmark to read:

“As a benchmark against which to compare properties of naturally occurring heteropolymeric IDPs, we inserted homopolymeric dipeptide repeats into our FRET construct. Specifically, we chose glycine-serine (GS) repeats for benchmarking because (1) they lack hydrophobicity, charge, and aromaticity which makes them easy to express and highly soluble⁶, (2) they have been shown to lack local and long-range structural biases, instead behaving as expected for a random coil across the range of lengths studied in our work^{38,41}, and (3) they have been shown to behave as real-chain mimics of ideal Gaussian chains in aqueous solutions⁴¹⁻⁴³.”

Page 13: Elongated structures under high crowding can be indeed explained. Another explanation could be the binding of molecular chaperones.

We have now shown that the specific effect observed for E1A of expansion under increased crowding is likely a result of attractive interactions between the E1A sequence and the folded FP to which it is tethered to (see the manuscript and **Fig. 5**.) Nonetheless, we acknowledge that this kind of effect can be a result of other factors as well. This is now stated in the text:

“This type of expansion under increased crowding has been previously reported³⁹, and may be caused inside the cell by protein-protein interactions including chaperone binding⁴⁰ or post-translational modifications⁴¹.”

The authors use solvent perturbation as a method to study the structural biases of IDPs in cells. Temperature is another variable with high physiological relevance for IDPs, e.g. formation of

stress granules under heat stress. In particular in the cell, temperature can be easily (and rapidly, e.g. by temperature jump) modulated. Have the authors conducted experiments under different incubation temperatures in cells?

This is an excellent question. Our past work has indeed shown that disordered and unfolded structures are significantly more sensitive to temperature changes inside the cell compared to well-folded constructs⁴². While this experiment is outside the scope of this work, we have now added some discussion of this aspect:

“This sensing capability of IDPs has been demonstrated not only for changes in solution conditions and osmotic pressure as studied here, but also for changes in other conditions such as membrane curvature⁴³, water availability⁴⁴, and temperature⁴⁵.”

Minor: Page 4: “Folded proteins are often compared to crystal structure”. I guess the authors want to say something like “The structure of folded proteins is often assumed to be identical to crystal structure ...?”

We thank the reviewer for this comment and have revised the text to read:

“The structure of a folded protein is commonly described in terms of its “native” conformation discerned through X-ray crystallography. For an IDP, no such single structure can be obtained. Instead, IDP structure is often described with reference to well-established homopolymer models^{46,47}.”

Reviewer references

1. Wicky, B. I. M., Shammash, S. L. & Clarke, J. Affinity of IDPs to their targets is modulated by ion-specific changes in kinetics and residual structure. *Proc. Natl. Acad. Sci. U. S. A.* **114**, 9882–9887 (2017).
2. Borchers, W. *et al.* Disorder and residual helicity alter p53-Mdm2 binding affinity and signaling in cells. *Nat. Chem. Biol.* **10**, 1000–1002 (2014).
3. Murthy, A. C. *et al.* Molecular interactions underlying liquid-liquid phase separation of the FUS low-complexity domain. *Nat. Struct. Mol. Biol.* **26**, 637–648 (2019).
4. Martin, E. W. *et al.* Sequence Determinants of the Conformational Properties of an Intrinsically Disordered Protein Prior to and upon Multisite Phosphorylation. *J. Am. Chem. Soc.* **138**, 15323–15335 (2016).
5. González-Foutel, N. S. *et al.* Conformational buffering underlies functional selection in intrinsically disordered protein regions. *Nat. Struct. Mol. Biol.* **29**, 781–790 (2022).
6. Patel, A. *et al.* A liquid-to-solid phase transition of the ALS protein FUS accelerated by disease mutation. *Cell* **162**, 1066–1077 (2015).
7. Theillet, F.-X. *et al.* Physicochemical properties of cells and their effects on intrinsically disordered proteins (IDPs). *Chem. Rev.* **114**, 6661–6714 (2014).
8. Kato, M. *et al.* Cell-free formation of RNA granules: low complexity sequence domains form dynamic fibers within hydrogels. *Cell* **149**, 753–767 (2012).
9. Baker, N. A., Sept, D., Joseph, S., Holst, M. J. & McCammon, J. A. Electrostatics of nanosystems: application to microtubules and the ribosome. *Proc. Natl. Acad. Sci. U. S. A.*

- 98**, 10037–10041 (2001).
10. Sievers, F. *et al.* Fast, scalable generation of high-quality protein multiple sequence alignments using Clustal Omega. *Mol. Syst. Biol.* **7**, 539 (2011).
 11. Goujon, M. *et al.* A new bioinformatics analysis tools framework at EMBL-EBI. *Nucleic Acids Res.* **38**, W695–9 (2010).
 12. Holehouse, A. S., Das, R. K., Ahad, J. N., Richardson, M. O. G. & Pappu, R. V. CIDER: Resources to Analyze Sequence-Ensemble Relationships of Intrinsically Disordered Proteins. *Biophys. J.* **112**, 16–21 (2017).
 13. Moses, D. *et al.* Revealing the Hidden Sensitivity of Intrinsically Disordered Proteins to their Chemical Environment. *J. Phys. Chem. Lett.* **11**, 10131–10136 (2020).
 14. Vancraenenbroeck, R., Harel, Y. S., Zheng, W. & Hofmann, H. Polymer effects modulate binding affinities in disordered proteins. *Proc. Natl. Acad. Sci. U. S. A.* (2019) doi:10.1073/pnas.1904997116.
 15. Pegram, L. M. & Record, M. T., Jr. Thermodynamic origin of Hofmeister ion effects. *J. Phys. Chem. B* **112**, 9428–9436 (2008).
 16. Song, J., Gomes, G.-N., Shi, T., Gradinaru, C. C. & Chan, H. S. Conformational Heterogeneity and FRET Data Interpretation for Dimensions of Unfolded Proteins. *Biophys. J.* **113**, 1012–1024 (2017).
 17. Peran, I. *et al.* Unfolded states under folding conditions accommodate sequence-specific conformational preferences with random coil-like dimensions. *Proc. Natl. Acad. Sci. U. S. A.* **116**, 12301–12310 (2019).
 18. Gomes, G.-N. W. *et al.* Conformational Ensembles of an Intrinsically Disordered Protein Consistent with NMR, SAXS, and Single-Molecule FRET. *J. Am. Chem. Soc.* **142**, 15697–15710 (2020).
 19. Ruff, K. M. & Holehouse, A. S. SAXS versus FRET: A Matter of Heterogeneity? *Biophys. J.* (2017) doi:10.1016/j.bpj.2017.07.024.
 20. Auton, M. & Bolen, D. W. Predicting the energetics of osmolyte-induced protein folding/unfolding. *Proc. Natl. Acad. Sci. U. S. A.* **102**, 15065–15068 (2005).
 21. Santoro, M. M., Liu, Y., Khan, S. M., Hou, L. X. & Bolen, D. W. Increased thermal stability of proteins in the presence of naturally occurring osmolytes. *Biochemistry* **31**, 5278–5283 (1992).
 22. Yancey, P. H., Clark, M. E., Hand, S. C., Bowlus, R. D. & Somero, G. N. Living with water stress: evolution of osmolyte systems. *Science* **217**, 1214–1222 (1982).
 23. Knowles, D. B. *et al.* Chemical Interactions of Polyethylene Glycols (PEGs) and Glycerol with Protein Functional Groups: Applications to Effects of PEG and Glycerol on Protein Processes. *Biochemistry* **54**, 3528–3542 (2015).
 24. Sukenik, S., Sapir, L. & Harries, D. Balance of enthalpy and entropy in depletion forces. *Curr. Opin. Colloid Interface Sci.* (2013).
 25. Qian, D. *et al.* Tie-lines reveal interactions driving heteromolecular condensate formation. *bioRxiv* 2022.02.22.481401 (2022) doi:10.1101/2022.02.22.481401.
 26. André, A. A. M., Yewdall, N. A. & Spruijt, E. Crowding-induced phase separation and gelling by co-condensation of PEG in NPM1-rRNA condensates. *Biophys. J.* **122**, 397–407 (2023).
 27. Mastop, M. *et al.* Characterization of a spectrally diverse set of fluorescent proteins as FRET acceptors for mTurquoise2. *Sci. Rep.* **7**, 11999 (2017).
 28. Sukenik, S., Ren, P. & Gruebele, M. Weak protein-protein interactions in live cells are quantified by cell-volume modulation. *Proc. Natl. Acad. Sci. U. S. A.* **114**, 6776–6781 (2017).
 29. Cuevas-Velazquez, C. L. *et al.* Intrinsically disordered protein biosensor tracks the physical-chemical effects of osmotic stress on cells. *Nat. Commun.* **12**, 5438 (2021).
 30. Shaner, N. C. *et al.* A bright monomeric green fluorescent protein derived from *Branchiostoma lanceolatum*. *Nat. Methods* **10**, 407–409 (2013).

31. Balleza, E., Kim, J. M. & Cluzel, P. Systematic characterization of maturation time of fluorescent proteins in living cells. *Nat. Methods* **15**, 47–51 (2018).
32. Chen, X., Zaro, J. L. & Shen, W.-C. Fusion protein linkers: property, design and functionality. *Adv. Drug Deliv. Rev.* **65**, 1357–1369 (2013).
33. Sørensen, C. S. & Kjaergaard, M. Effective concentrations enforced by intrinsically disordered linkers are governed by polymer physics. *Proc. Natl. Acad. Sci. U. S. A.* **116**, 23124–23131 (2019).
34. Sørensen, C. S. & Kjaergaard, M. Measuring Effective Concentrations Enforced by Intrinsically Disordered Linkers. *Methods Mol. Biol.* **2141**, 505–518 (2020).
35. Basak, S. *et al.* Probing Interdomain Linkers and Protein Supertertiary Structure In Vitro and in Live Cells with Fluorescent Protein Resonance Energy Transfer. *J. Mol. Biol.* **433**, 166793 (2021).
36. Das, R. K., Ruff, K. M. & Pappu, R. V. Relating sequence encoded information to form and function of intrinsically disordered proteins. *Curr. Opin. Struct. Biol.* **32**, 102–112 (2015).
37. Möglich, A., Joder, K. & Kiefhaber, T. End-to-end distance distributions and intrachain diffusion constants in unfolded polypeptide chains indicate intramolecular hydrogen bond formation. *Proc. Natl. Acad. Sci. U. S. A.* **103**, 12394–12399 (2006).
38. Evers, T. H., Van Dongen, E. M. W. M., Faesen, A. C., Meijer, E. W. & Merkx, M. Quantitative understanding of the energy transfer between fluorescent proteins connected via flexible peptide linkers. *Biochemistry* **45**, 13183–13192 (2006).
39. Banks, A., Qin, S., Weiss, K. L., Stanley, C. B. & Zhou, H. X. Intrinsically Disordered Protein Exhibits Both Compaction and Expansion under Macromolecular Crowding. *Biophys. J.* **114**, 1067–1079 (2018).
40. Tsvetkov, P., Reuven, N. & Shaul, Y. The nanny model for IDPs. *Nat. Chem. Biol.* **5**, 778–781 (2009).
41. Bah, A. & Forman-Kay, J. D. Modulation of intrinsically disordered protein function by post-translational modifications. *J. Biol. Chem.* **291**, 6696–6705 (2016).
42. Wang, Y., Sukenik, S., Davis, C. M. & Gruebele, M. Cell Volume Controls Protein Stability and Compactness of the Unfolded State. *J. Phys. Chem. B* **122**, 11762–11770 (2018).
43. Zeno, W. F. *et al.* Molecular Mechanisms of Membrane Curvature Sensing by a Disordered Protein. *J. Am. Chem. Soc.* **141**, 10361–10371 (2019).
44. Cuevas-Velazquez, C. L., Saab-Rincón, G., Reyes, J. L. & Covarrubias, A. A. The Unstructured N-terminal Region of Arabidopsis Group 4 Late Embryogenesis Abundant (LEA) Proteins Is Required for Folding and for Chaperone-like Activity under Water Deficit. *J. Biol. Chem.* **291**, 10893–10903 (2016).
45. Wuttke, R. *et al.* Temperature-dependent solvation modulates the dimensions of disordered proteins. *Proc. Natl. Acad. Sci. U. S. A.* **111**, 5213–5218 (2014).
46. Hofmann, H. *et al.* Polymer scaling laws of unfolded and intrinsically disordered proteins quantified with single-molecule spectroscopy. *Proceedings of the National Academy of Sciences* **109**, 16155–16160 (2012).
47. Soranno, A. *et al.* Single-molecule spectroscopy reveals polymer effects of disordered proteins in crowded environments. *Proc. Natl. Acad. Sci. U. S. A.* **111**, 4874–4879 (2014).

Decision Letter, first revision:**Message:** 11th May 2023

Dear Dr. Sukenik,

Thank you again for submitting your manuscript "Structural biases in disordered proteins are prevalent in the cell". I apologize for the delay in responding, which resulted from the difficulty in obtaining the referee reports. Nevertheless, we now have comments (below) from the 2 reviewers who evaluated your paper. In light of those reports, we remain interested in your study and would like to see your response to the comments of the referees, in the form of a revised manuscript, before we can make a final decision.

You will see that while both reviewers appreciate how improved the revised manuscript is, Reviewer #1 points out missing critical controls that question the key message of the paper. Please be sure to address/respond to all concerns of the referees in full in a point-by-point response and highlight all changes in the revised manuscript text file. If you have comments that are intended for editors only, please include those in a separate cover letter.

Seeing as performing these experiments might require some time, we would expect to see your revised manuscript in 3-6 months. If you cannot send it within this time, please contact us to discuss an extension; we would still consider your revision, provided that no similar work has been accepted for publication at NSMB or published elsewhere.

Reporting Summary:

When submitting the revised version of your manuscript, please pay close attention to our [href="https://www.nature.com/nature-portfolio/editorial-policies/image-integrity">Digital Image Integrity Guidelines.](https://www.nature.com/nature-portfolio/editorial-policies/image-integrity) and to the following points below:

-- that unprocessed scans are clearly labelled and match the gels and western blots

presented in figures.

-- that control panels for gels and western blots are appropriately described as loading on sample processing controls

-- all images in the paper are checked for duplication of panels and for splicing of gel lanes.

Please note that all key data shown in the main figures as cropped gels or blots should be presented in uncropped form, with molecular weight markers. These data can be aggregated into a single supplementary figure item. While these data can be displayed in a relatively informal style, they must refer back to the relevant figures. These data should be submitted with the final revision, as source data, prior to acceptance, but you may want to start putting it together at this point.

SOURCE DATA: we request that authors to provide, in tabular form, the data underlying the graphical representations used in figures. This is to further increase transparency in data reporting, as detailed in this editorial (<http://www.nature.com/nsmb/journal/v22/n10/full/nsmb.3110.html>). Spreadsheets can be submitted in excel format. Only one (1) file per figure is permitted; thus, for multi-paneled figures, the source data for each panel should be clearly labeled in the Excel file; alternately the data can be provided as multiple, clearly labeled sheets in an Excel file. When submitting files, the title field should indicate which figure the source data pertains to. We encourage our authors to provide source data at the revision stage, so that they are part of the peer-review process.

Data availability: this journal strongly supports public availability of data. All data used in accepted papers should be available via a public data repository, or alternatively, as Supplementary Information. If data can only be shared on request, please explain why in your Data Availability Statement, and also in the correspondence with your editor. Please note that for some data types, deposition in a public repository is mandatory - more information on our data deposition policies and available repositories can be found below: <https://www.nature.com/nature-research/editorial-policies/reporting-standards#availability-of-data>

While we encourage the use of color in preparing figures, please note that this will incur a charge to partially defray the cost of printing. Information about color charges can be

found at <http://www.nature.com/nsmb/authors/submit/index.html#costs>

[redacted]

Sincerely,
Sara

Sara Osman, Ph.D.
Associate Editor
Nature Structural & Molecular Biology

Reviewers' Comments:

Reviewer #1:

Remarks to the Author:

The authors did an excellent job in revising the manuscript and addressed most of my concerns.

However, the added new experiments with flipped FPs seem to confirm some of my initial concerns regarding the impact of the selected FPs. As the authors state, all reported differences are statistically significant. That means that flipping FPs leads to results that are statistically significantly different. More importantly, the new results bring up the question whether the observed correlations between the ensembles in vitro and in cells are due to a dominant effect of FPs on the system. In other words, do most systems (not all, e.g. p53) behave in a similar manner in vitro and in cells, and in a specific manner under changing environments, because of specific IDP-FP interactions that dominate the measured signal (e.g. FRET or SAXS). The authors state: "... IDP ensembles are able to sense and respond to changes in the composition of their environment." The added controls suggest that it is hard to disentangle what of the measured changes can be attributed truly to properties intrinsic to the IDPs and what originates from IDP-FP interactions. In other words, are measured changes induced by a different environment

mainly caused by IDP-FP interaction changes or changes in IDP behaviour?

As these concerns question the key message of the paper, additional controls are necessary. For instance, the authors could provide some additional controls comparing the impact of flipped FPs in vitro and in cells. E.g. the equivalent of 5D in vitro.

Reviewer #2:

Remarks to the Author:

The authors fully addressed my comments and concerns and the manuscript was significantly improved. I recommend publication.

Author Rebuttal, first revision:

Reviewer #1

The authors did an excellent job in revising the manuscript and addressed most of my concerns. However, the added new experiments with flipped FPs seem to confirm some of my initial concerns regarding the impact of the selected FPs. As the authors state, all reported differences are statistically significant. That means that flipping FPs leads to results that are statistically significantly different. More importantly, the new results bring up the question whether the observed correlations between the ensembles *in vitro* and in cells are due to a dominant effect of FPs on the system. In other words, do most systems (not all, e.g. p53) behave in a similar manner *in vitro* and in cells, and in a specific manner under changing environments, because of specific IDP-FP interactions that dominate the measured signal (e.g. FRET or SAXS). The authors state: "... IDP ensembles are able to sense and respond to changes in the composition of their environment." The added controls suggest that it is hard to disentangle what of the measured changes can be attributed truly to prosperities intrinsic to the IDPs and what originates from IDP-FP interactions. In other words, are measured changes induced by a different environment mainly caused by IDP-FP interaction changes or changes in IDP behaviour? As these concerns question the key message of the paper, additional controls are necessary. For instance, the authors could provide some additional controls comparing the impact of flipped FPs *in vitro* and in cells. E.g. the equivalent of 5D *in vitro*.

We thank the reviewer for their comments and suggestion of experiments to resolve some of their concerns. We have now performed the *in vitro* experiments suggested by the reviewer, and present the results (**Fig. R1**, left; **Fig. 5B** in the new revision) in comparison with the live-cell results provided in the first revision (**Fig. R1**, right; **Fig. 5C** in the new revision). The new experiments show that the *in vitro* results for naturally occurring IDPs match well with the live-cell results. In particular, E1A *in vitro* shows a difference in FRET efficiency between the original and flipped constructs that matches what was seen in cells, while p53 and Ash1 show little change in FRET efficiency between the original and flipped constructs both *in vitro* and in live cells.

Figure R1. New *in vitro* original vs. flipped results (left) match previous live-cell results (right).

We also looked at Stokes shifts as in **Fig. S17D**, and our results again imply that differences between IDP:FP interactions at the N and C terminals of mNeonGreen may account for the

measured differences in FRET between the original and flipped constructs (**Fig. R2**). In particular, the trends for p53 and Ash1 are similar to GS16, while the trends for E1A have a higher peak position for the original construct compared to the flipped one. This is a good example of the variability in interactions that may occur between IDRs and folded domains throughout nature, which clearly should be the subject of further study. However, this does not seem to pose a problem for our current study: The IDP:FP interactions observed *in vitro* are sequence-specific, and are recapitulated inside the cellular environment. Furthermore, in the cellular environment, changing these interactions will also change sensitivity to cellular conditions.

Figure R2. Peak wavelengths of mTurquoise2 and mNeonGreen untethered and as part of constructs with different IDRs.

While we show that for 2 out of 3 IDPs tested FRET results were similar between original and flipped constructs, these experiments do not alleviate the reviewer's concern – which is also the underlying cause of a long-standing controversy in the IDP literature¹⁻⁷ – that any label can and will interact with disordered regions and affect measured results. Our new results show that

FP:IDP interactions do occur, and at least to us it would be surprising if they were completely absent. The only way to ensure these interactions do not exist is to measure IDP ensemble dimensions without labels in live cells, a feat that to the best of our knowledge has not been achieved. However, even these experiments would be questionable as our results point to the importance of measuring IDPs in the context of their full-length protein: IDP:folded-domain interactions will likely occur in the full length sequence as well, in the same way as they occur with our exogenous FPs. Despite this, **we maintain that the existence of IDP:FP interactions in our dataset does not put into question the key conclusion of our paper.**

Our key conclusion, as stated in our title, is that the structural biases observed *in vitro* are also seen in live cells. This remains completely consistent throughout our entire dataset, regardless of whether these structural biases contain interactions with FP labels or not. Our new results show that, at the very least, there is a strong sequence dependence in determining the breakdown of IDP:IDP interactions vs. IDP:folded-domain interactions in shaping full-length IDP ensembles. This is true whether this folded domain is exogenous, as is the case for the FPs used here, or endogenous as is the case for > 90% of IDPs in the human proteome⁸. Thus, the experiments proposed by the reviewer have helped us to draw a more general conclusion than we could have drawn without those experiments.

In our revised version, beyond the new data and analyses, we emphasized that our definition of “IDP structural biases” includes both IDP:IDP interactions and IDP:folded-domain interactions. We have highlighted in yellow paragraphs that show this in the annotated revised manuscript. Furthermore, we have added the following paragraph discussing IDP:FP interactions in our “Limitations and drawbacks” section:

“As a final point, we acknowledge that interactions between the studied IDPs and the FPs that make up our FRET construct exist and likely affect the dimensions of our measured ensembles. To address this, we point to the fact that nearly all studied IDPs (including those in this work) are excised from full-length proteins in which they would be tethered to folded domains. Our results point to the importance of the intramolecular context of an IDP: interactions with a tethered folded domain can alter IDP ensembles, as well as their response to changes in the cell. The importance of IDP:folded domain interactions has already been pointed out in several recent studies^{8–11}. Despite all this, our results show that even where FP:IDP interactions exist, the structural biases shaping disordered protein ensembles *in vitro* are recapitulated in the cell.”

We hope that these new experiments, additional analyses, and expanded discussion in the text help address the reviewer’s concerns regarding the role of IDP:FP interactions in determining the observed ensembles, and that the paper can now be accepted for publication.

REFERENCES

1. Riback, J. A. *et al.* Innovative scattering analysis shows that hydrophobic disordered proteins are expanded in water. *Science* **358**, 238–241 (2017).

2. Riback, J. A. *et al.* Commonly used FRET fluorophores promote collapse of an otherwise disordered protein. *Proceedings of the National Academy of Sciences* 201813038 (2019).
3. Song, J., Gomes, G.-N., Shi, T., Gradinaru, C. C. & Chan, H. S. Conformational Heterogeneity and FRET Data Interpretation for Dimensions of Unfolded Proteins. *Biophys. J.* **113**, 1012–1024 (2017).
4. Best, R. B. *et al.* Comment on ‘Innovative scattering analysis shows that hydrophobic disordered proteins are expanded in water’. *Science* vol. 361 (2018).
5. Riback, J. A. *et al.* Response to Comment on ‘Innovative scattering analysis shows that hydrophobic disordered proteins are expanded in water’. *Science* vol. 361 (2018).
6. Gomes, G.-N. W. *et al.* Conformational Ensembles of an Intrinsically Disordered Protein Consistent with NMR, SAXS, and Single-Molecule FRET. *J. Am. Chem. Soc.* **142**, 15697–15710 (2020).
7. Ruff, K. M. & Holehouse, A. S. SAXS versus FRET: A Matter of Heterogeneity? *Biophys. J.* (2017) doi:10.1016/j.bpj.2017.07.024.
8. Taneja, I. & Holehouse, A. S. Folded domain charge properties influence the conformational behavior of disordered tails. *Curr Res Struct Biol* **3**, 216–228 (2021).
9. Mittal, A., Holehouse, A. S., Cohan, M. C. & Pappu, R. V. Sequence-to-Conformation Relationships of Disordered Regions Tethered to Folded Domains of Proteins. *J. Mol. Biol.* (2018) doi:10.1016/j.jmb.2018.05.012.
10. Martin, E. W. *et al.* Interplay of folded domains and the disordered low-complexity domain in mediating hnRNPA1 phase separation. *Nucleic Acids Res.* **49**, 2931–2945 (2021).
11. Zheng, T., Galagedera, S. K. K. & Castañeda, C. A. Previously uncharacterized interactions between the folded and intrinsically disordered domains impart asymmetric effects on UBQLN2 phase separation. *Protein Sci.* **30**, 1467–1481 (2021).

Decision Letter, second revision:

Message: Our ref: NSMB-A46350B

14th Jul 2023

Dear Dr. Sukenik,

Thank you for submitting your revised manuscript "Structural biases in disordered proteins are prevalent in the cell" (NSMB-A46350B). I apologize for the delay in responding, due to absences in our editorial team. Your manuscript has now been seen by the original Reviewer #1 and their comments are below. The reviewer finds that the paper has improved in revision, and therefore we'll be happy in principle to publish it in Nature Structural & Molecular Biology, pending minor revisions to comply with our editorial and formatting guidelines.

We are now performing detailed checks on your paper and will send you a checklist detailing our editorial and formatting requirements in a couple of weeks. Please do not upload the final materials and make any revisions until you receive this additional information from us.

To facilitate our work at this stage, it is important that we have a copy of the main text as a word file. If you could please send along a word version of this file as soon as possible, we would greatly appreciate it; please make sure to copy the NSMB account (cc'ed above).

Sincerely,
Sara

Sara Osman, Ph.D.
Associate Editor
Nature Structural & Molecular Biology

Reviewer #1 (Remarks to the Author):

The authors addressed all my concerns.

Final Decision Letter:

Message 4th Oct 2023

:

Dear Dr. Sukenik,

We are now happy to accept your revised paper "Structural biases in disordered proteins are prevalent in the cell" for publication as an Article in Nature Structural & Molecular Biology.

As soon as your article is published, you can generate your shareable link by entering the DOI of your article here: http://authors.springernature.com/share. Corresponding authors will also receive an automated email with the shareable link

Your paper will be published online soon after we receive proof corrections and will appear in print in the next available issue. You can find out your date of online publication by contacting the production team shortly after sending your proof corrections. Content is published online weekly on Mondays and Thursdays, and the embargo is set at 16:00 London time (GMT)/11:00 am US Eastern time (EST) on the day of publication. Now is the time to inform your Public Relations or Press Office about your paper, as they might be interested in promoting its publication. This will allow them time to prepare an accurate and satisfactory press release. Include your manuscript tracking number (NSMB-A46350C)

and our journal name, which they will need when they contact our press office.

About one week before your paper is published online, we shall be distributing a press release to news organizations worldwide, which may very well include details of your work. We are happy for your institution or funding agency to prepare its own press release, but it must mention the embargo date and Nature Structural & Molecular Biology. If you or your Press Office have any enquiries in the meantime, please contact press@nature.com.

Please note that *Nature Structural & Molecular Biology* is a Transformative Journal (TJ). Authors may publish their research with us through the traditional subscription access route or make their paper immediately open access through payment of an article-processing charge (APC). Authors will not be required to make a final decision about access to their article until it has been accepted. <https://www.springernature.com/gp/open-research/transformative-journals> Find out more about Transformative Journals

Sincerely,
Sara

Sara Osman, Ph.D.
Associate Editor
Nature Structural & Molecular Biology
